# Fusing Pixels and Genes: Spatially-Aware Learning in Computational Pathology

**Minghao Han[1,2], Dingkang Yang[1,2,*], Linhao Qu[3], Zizhi Chen[1,2], Gang Li[4], Han Wang[5], Jiacong Wang[5], Lihua Zhang[1,2,*]**

[1]College of Intelligent Robotics and Advanced Manufacturing, Fudan University,
[2]Fysics Intelligence Technologies Co., Ltd. (Fysics AI),
[3]Department of Biomedical Informatics, Harvard Medical School,
[4]Tencent Youtu Lab, [5]ByteDance
{mhhan22@m.fudan.edu.cn, dicken@fyscis.ai, lihuazhang@fudan.edu.cn}

## ABSTRACT

Recent years have witnessed remarkable progress in multimodal learning within computational pathology. Existing models primarily rely on vision and language modalities; however, language alone lacks molecular specificity and offers limited pathological supervision, leading to representational bottlenecks. In this paper, we propose STAMP, a **S**patial **T**ranscriptomics-**A**ugmented **M**ultimodal **P**athology representation learning framework that integrates spatially-resolved gene expression profiles to enable molecule-guided joint embedding of pathology images and transcriptomic data. Our study shows that self-supervised, gene-guided training provides a robust and task-agnostic signal for learning pathology image representations. Incorporating spatial context and multi-scale information further enhances model performance and generalizability. To support this, we constructed SpaVis-6M, the largest Visium-based spatial transcriptomics dataset to date, and trained a spatially-aware gene encoder on this resource. Leveraging hierarchical multi-scale contrastive alignment and cross-scale patch localization mechanisms, STAMP effectively aligns spatial transcriptomics with pathology images, capturing spatial structure and molecular variation. We validate STAMP across six datasets and four downstream tasks, where it consistently achieves strong performance. These results highlight the value and necessity of integrating spatially resolved molecular supervision for advancing multimodal learning in computational pathology. The code is included in the supplementary materials. The pretrained weights and SpaVis-6M are available at: https://github.com/Hanminghao/STAMP.

## 1 INTRODUCTION

In recent years, computational pathology (CPATH) has made significant progress in multiple cancer-related downstream tasks using deep learning techniques (Song et al., 2024; Wang et al., 2024b;a; Shi et al., 2023). However, existing methods are often designed for specific datasets or tasks, potentially leading to performance degradation when models encounter distribution shifts in new datasets or tasks. An effective strategy to address this challenge is to collect large-scale cross-domain data and train foundation models capable of adapting to diverse scenarios. Some researchers propose that developing foundation models that generalize to a wide range of downstream tasks is more cost-effective and scientifically robust than investing substantial effort in designing complex, task-specific downstream models (Chen et al., 2024b; Lu et al., 2024; Ikezogwo et al., 2023; Han et al., 2025a).

Recent studies have shown that large-scale multimodal pretraining using noisy image-text pairs can improve downstream task performance (Radford et al., 2021; Li et al., 2022). Inspired by this, various contrastive learning frameworks have been proposed in the CPATH domain, such as QuiltNet (Ikezogwo et al., 2023) and CONCH (Lu et al., 2024), which pretrain models using paired pathology images and descriptive text. Although natural language is widely used in pathology

---
*Corresponding authors.

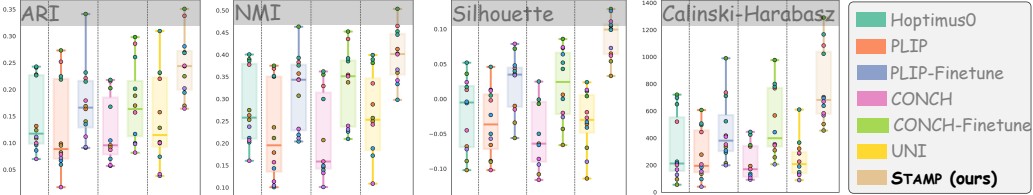

Figure 1: Gene-guided supervision boosts vision encoders. We evaluated unsupervised clustering on the DLPFC dataset. Models fine-tuned with spatial transcriptomics supervision and STAMP consistently outperform baselines across four metrics (ARI, NMI, Silhouette, and Calinski-Harabasz), demonstrating that molecular information enhances biological structure identification.

analysis, multimodal pretraining based on image-text pairs cannot provide additional diagnostic information and struggles to reveal key molecular mechanisms crucial for cancer research. In contrast, gene expression data can provide complementary information at the molecular level, aiding in deciphering carcinogenic mechanisms and supporting personalized treatment strategies (Xu et al., 2024c; Ding et al., 2023). To address this limitation, TANGLE (Jaume et al., 2024b) proposed using bulk RNA to guide representation learning of Whole Slide Images (WSIs), significantly improving downstream task performance. However, this method relies on bulk RNA data and can only capture patient-level information, neglecting intra-sample heterogeneity (Li & Wang, 2021).

Unlike bulk RNA sequencing, spatial transcriptomics (ST) combines pathology images with RNA expression analysis, allowing localization and quantification of RNA within WSIs. This technique bridges the gap between pathology images and gene expression data (Jain & Eadon, 2024). In ST technology, H&E-stained WSIs are fixed on chips containing multiple capture spots, with each spot collecting mRNA from multiple cells in its area. Through sequencing, pathology images are mapped to gene expression profiles, similar to the construction of image-text pairs. Pathology images capture tissue structure, cellular morphology, and tumor invasion (Qu et al., 2024; Han et al., 2025b), while gene expression data reveals the tumor microenvironment and underlying disease mechanisms (Schaar et al., 2024). Together, they offer complementary insights into cancer analysis. In addition, studies have shown a direct correlation between gene expression variation and morphological features, highlighting a deeper connection between the two modalities and further justifying gene-guided representation learning (Kueckelhaus et al., 2024). Following prior work (Chen et al., 2024a), we adapted PLIP (Huang et al., 2023) and CONCH (Lu et al., 2024) using ST-based supervision on the DLPFC dataset (Maynard et al., 2021). As shown in Figure 1, we evaluated this by performing unsupervised clustering on the DLPFC dataset and comparing the results against the seven ground-truth anatomical layers (L1-L6, WM). The fine-tuned models (PLIP-Finetune and CONCH-Finetune) achieve superior scores across all four clustering metrics: ARI (Steinley, 2004), NMI (Estévez et al., 2009), Silhouette (Rousseeuw, 1987), and Calinski-Harabasz (Caliński & Harabasz, 1974). This demonstrates that gene-guided supervision enables models to identify biological structures better.

Although spatial transcriptomic supervision has shown promise, existing approaches suffer from two key limitations. First, they remain **overly simplistic and poorly generalizable**: most methods encode only a subset of genes via basic linear layers and require full-parameter fine-tuning of vision backbones on each new dataset (Chen et al., 2024a). Second, they **ignore the inherently spatial, multi-scale nature** of ST data. Unlike independent image-text pairs, ST measurements and their corresponding image patches co-exist within the same tissue, exhibiting strong spatial dependencies across neighboring capture spots (Wang et al., 2025b). Directly importing vision-language pretraining techniques, therefore, fails to leverage ST's rich spatial context and hierarchical features (Han et al., 2024b). Thus, developing a large-scale, spatially aware multimodal pretraining framework for pathology images and spatial transcriptomics is key to advancing computational pathology.

To address these limitations, we propose the **S**patial **T**ranscriptomics-**A**ugmented **M**ultimodal **P**athology (STAMP) representation learning framework, a large-scale, generalizable multimodal pretraining approach for joint representation learning of pathology images and spatial transcriptomics gene expression profiles. Specifically, when training the spatial-aware gene encoder (first stage in §3.2), STAMP employs a spatial sampling strategy and a new neighborhood training objective to model over 5.75 million spatial transcriptomics entries, effectively capturing spatial co-localization

patterns among spot states within tissues. Furthermore, in the alignment pretraining stage (second stage in §3.3), using 697K pairs of pathology images and spatial transcriptomics data, STAMP enhances the vision encoder's ability to perceive spatial relationships and multi-scale features through hierarchical multi-scale contrastive alignment and cross-scale localization mechanisms for pathology images. The main contributions of this paper are as follows:

• We present the Spatial Visium Transcriptomics Dataset (SpaVis-6M), currently the largest spatial transcriptomics dataset based on 10X Visium technology. SpaVis-6M consists of 5.75 million spatial transcriptomics gene expression entries from 35 organs, 1,982 slices, and 262 datasets or publications. This provides strong support for training the robust spatial-aware gene encoder.

• We propose the **S**patial **T**ranscriptomics-**A**ugmented **M**ultimodal **P**athology representation learning framework (STAMP), trained on 5.75 million spatial transcriptomics entries and 697K pathology image-gene expression pairs. To the best of our knowledge, STAMP is among the first large-scale frameworks designed for multimodal representation learning in pathology images and ST data.

• We propose a unified alignment loss for STAMP that synergistically combines specialized objectives (targeting spatial localization, inter-modal feature matching, and intra-modal multi-scale consistency) to capture spatial structures and molecular variations effectively.

• We conduct experiments on six datasets and four downstream tasks, where STAMP achieved state-of-the-art (SOTA) performance, demonstrating the powerful performance and necessity of multimodal pretraining using gene expression data as a supervisory signal.

## 2 RELATED WORK

**Foundation Models in Computational Pathology and Spatial Transcriptomics.** Early pretraining for pathology foundation models demonstrated the potential of self-supervised learning on large-scale image datasets using techniques like MoCo (Chen et al., 2021) and masked image modeling (He et al., 2022; Chen et al., 2024b; Saillard et al., 2024). However, as single-modality methods struggle to capture complex disease biology (Lu et al., 2024), the field moved towards multimodal approaches. The first step involved aligning pathology images with natural language, with models like PLIP (Huang et al., 2023), QuiltNet (Ikezogwo et al., 2023), and CONCH (Lu et al., 2024). While useful, language lacks the molecular depth crucial for precision medicine (Li & Wang, 2021; Oksza-Orzechowski et al., 2024). To incorporate deeper molecular supervision, pioneering works like TANGLE (Jaume et al., 2024b) and mSTAR (Xu et al., 2024c) began using bulk RNA-seq data to guide Whole Slide Image (WSI) representation learning. A key limitation, however, is that bulk RNA averages gene expression across the entire tissue, overlooking the critical intra-sample spatial heterogeneity (Chu et al., 2022). Spatial transcriptomics (ST) provides an effective method to address this problem and preserve spatial context. scGPT (Cui et al., 2024) first validated the effectiveness of the Transformer architecture on massive single-cell datasets. Subsequently, models like Nicheformer (Schaar et al., 2024) and scGPT-Spatial (Wang et al., 2025a) further introduced spatial information. However, these ST foundation models are predominantly trained on single-cell or subcellular level data and rarely include ST data that can be directly matched with corresponding high-resolution pathology images. Distinct from all prior approaches, our work aims to bridge this gap by leveraging spatial transcriptomics as a supervisory signal for spatially-aware multimodal pretraining, aligning pathology images with gene expression profiles to capture tissue-level heterogeneity and enhance model performance.

**Spatial Transcriptomics in Computational Pathology.** Spatial transcriptomics is a revolutionary technology that combines pathology imaging with RNA expression analysis, enabling the localization and quantification of RNA within the spatial context of tissue sections (Jain & Eadon, 2024). Unlike bulk RNA-seq (Chen et al., 2025b), spatial transcriptomics preserves the spatial organization of gene expression patterns, providing crucial insights into tumor microenvironments and cell-cell interactions driving disease progression. Early applications of spatial transcriptomics in computational pathology mainly focused on regression-based prediction of gene expression from pathology images using supervised learning methods (He et al., 2020; Yang et al., 2023; Zeng et al., 2022; Chung et al., 2024). Notably, HisToGene (Pang et al., 2021) introduced Vision Transformers to explicitly model the spatial dependency of measured spots for accurate prediction. In recent years, methods such as BLEEP (Xie et al., 2024) and mclSTExp (Min et al., 2024) have explored contrastive learning frameworks to align these two modalities, followed by query-reference strategies for gene expression prediction. Building on this, OmiCLIP (Chen et al., 2025a) aligns tissue patches with transcriptomic

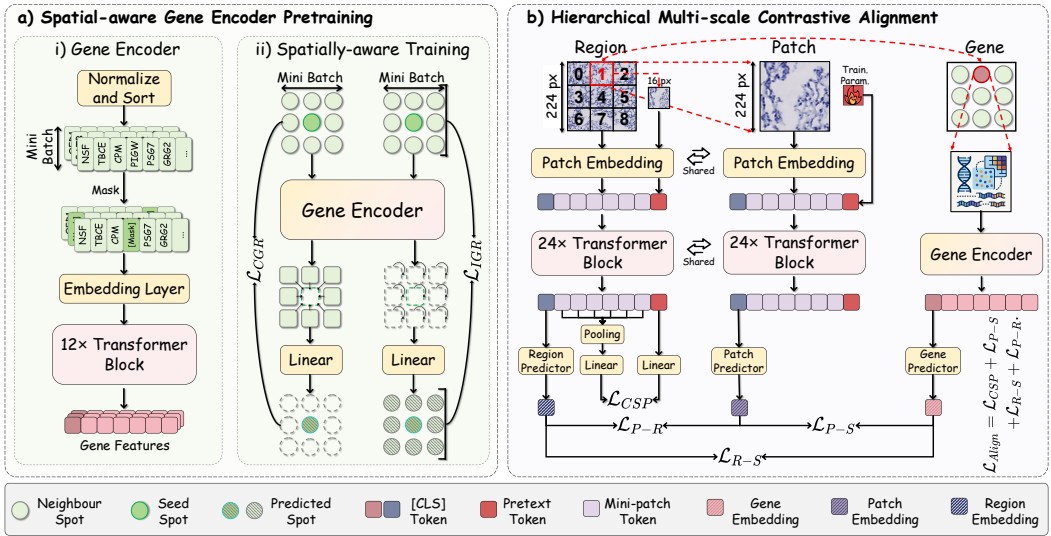

Figure 2: **Overview of STAMP's Two-Stage Pretraining Framework.** The framework is divided into two stages: **(a) Spatial-aware Gene Encoder Pretraining** uses 5.75 million spatial transcriptomics gene expression data to pretrain the gene encoder. **(b) Hierarchical Multi-scale Contrastive Alignment** adopts a pretrained pathological vision transformer as the vision encoder, aligning it with the gene encoder via hierarchical contrastive learning to fuse the two modalities.

data by transforming gene signatures into textual sentences, whereas UMPIRE (Han et al., 2024a) proposes a unified framework to fuse pathology images with gene expression profiles via multimodal training. However, these attempts were often limited by data scale or simple feature interactions, potentially diminishing generalizability. Large-scale cross-domain data is an effective method to mitigate batch effects and enhance model robustness. Still, spatial transcriptomics experiments are costly and time-consuming, resulting in relatively small available datasets (Jain & Eadon, 2024). In response to this challenge, we pioneered the construction of SpaVis-6M, an unprecedentedly large spatial transcriptomics dataset, and leveraged a two-stage pretraining strategy.

## 3 METHODOLOGY

Figure 2 presents the core of the STAMP pipeline: its two-stage pretraining process along with the key architectural components involved. STAMP is trained on the largest 10X Visium-based spatial transcriptomics (ST) dataset (*i.e.*, SpaVis-6M) and the largest pathology image-spatial transcriptomics paired dataset (*i.e.*, HEST (Jaume et al., 2024a)). Although HEST is the largest dataset in this field, only 697K data pairs from 329 slides remain after filtering, which is significantly fewer than datasets in vision-language models (*e.g.*, CONCH (Lu et al., 2024) with 1.17M pairs). To lessen its dependence on paired data, STAMP utilizes a deliberately designed two-stage pretraining strategy, including: **Spatial-aware Gene Encoder Pretraining** and **Hierarchical Multi-scale Contrastive Alignment**.

### 3.1 SPAVIS-6M: THE LARGEST VISIUM-BASED SPATIAL TRANSCRIPTOMICS DATASET

**Advantages of 10X Visium Data.** We choose the 10X Visium platform (10x Genomics, Inc., 2019) for training on ST gene expression data due to its key advantages: (1) a 55-micrometer resolution that matches typical tissue section dimensions (Jain & Eadon, 2024), (2) the ability to capture a broader range of genes compared to other platforms (Jain & Eadon, 2024), and (3) the abundance and accessibility of 10X Visium datasets (Jaume et al., 2024a; Chen et al., 2024a). Although trained exclusively on 10X Visium data, our model generalizes well to other sequencing platforms.

**Composition of SpaVis-6M.** To enhance cross-domain generalization in gene encoding, we developed SpaVis-6M, the largest 10X Visium-based ST dataset. This comprehensive resource provides unprecedented scale and diversity for pretraining. As illustrated in Figure 4, SpaVis-6M encompasses

1982 slices derived from 35 distinct organs and 262 studies/datasets. This collection comprises 5.75 million spatial transcriptomic gene expression profiles. See Appendix A.2.2 for details.

## 3.2 Spatial-aware Gene Encoder Pretraining

Unlike foundation models in pathology imaging and single-cell, ST remains in its early stages. Existing ST foundation models primarily focus on single-cell-level techniques, which lack corresponding pathology images (Schaar et al., 2024; Wang et al., 2025a). Moreover, these models treat ST data as single-cell data during pretraining, failing to incorporate the spatially structured information (Schaar et al., 2024). To address this limitation, as shown in Figure 2a, we propose leveraging the SpaVis-6M dataset to pretrain a spatially aware gene encoder specifically for Visium data.

**Gene Vocabulary.** SpaVis-6M and HEST (Jaume et al., 2024a) are composed of multiple datasets, with varying experimental standards leading to differences in gene nomenclature. To avoid redundancy or confusion, we standardize gene names using **Ensembl IDs**. Additionally, the number of measured genes varies across datasets. For consistency, we select 20,310 genes, including human protein-coding genes and mitochondrial genes, with missing values imputed as zero.

**Gene Tokenization.** Identifying and ranking genes with expression levels that significantly deviate from the mean has proven effective in capturing tissue-specific structures and disease-related mechanisms (Arora et al., 2023; Lerma-Martin et al., 2024). Consequently, we employ an **abnormal gene expression ranking** strategy to represent overarching gene information. As shown in Algorithm 1, we first compute the average expression level of each gene across all samples with nonzero expression. Then, to mitigate batch effects, we normalize each gene's expression by dividing it by its corresponding average expression value. Critically, instead of using these normalized values directly, which can be distorted by batch effects and hinder model training, we tokenize by sorting these standardized deviations in descending order. Like BERT's sequence ordering, this rank-based tokenization offers a robust representation of gene activity less susceptible to inter-batch variations. Mathematically, the resulting token sequence $T_i$ for a sample $i$ is formally defined as:

$$T_i = \{id(ep_i^0), id(ep_i^1), \ldots, id(ep_i^{N-1}) : ep_i^k \geq ep_i^{k+1}\}, \tag{1}$$

where $id(ep_i^k)$ and $ep_i^k$ represent the index of gene $k$ in the gene vocabulary and the normalized gene expression of sample $i$. We set the gene encoder's context length to 1,500 tokens ($N = 1,500$). Crucially, our rank-based tokenization (Algorithm 1) inherently handles the data sparsity mentioned in the "Gene Vocabulary" section. Genes "imputed as zero" (*i.e.*, not detected in a sample) will not be ranked among the top N abnormal genes and are thus never selected into the final token sequence $T_i$. This design avoids the batch effects and training instability associated with zero-imputation.

**Model Architecture.** The gene encoder begins by embedding a tokenized gene vector $T_i \in \mathbb{R}^N$:

$$x_{i,0} = \text{Embedding}(T_i) + \text{PosEmbedding}(P_i), \tag{2}$$

where $x_{i,0} \in \mathbb{R}^{N \times D}$ represents the embedded sequence, and $D$ is the embedding dimension. Here, Embedding is a standard, learnable token embedding layer that maps each gene ID in the sequence $T_i$ to a $D$-dimensional vector. The positional embedding $P_i$ encodes the position information of $T_i$. The gene encoder comprises 12 transformer blocks ($L = 12$). Given the embedded sequence $x_{i,0} = x_i$, each transformer block updates the representation iteratively as follows:

$$x_{i,l+1} = \text{TransformerBlock}(x_{i,l}), \quad l \in [0, L-1]. \tag{3}$$

**Constructing Spatially Coherent Mini-Batches.** Standard training procedures, which treat ST spots as independent samples, disregard the crucial spatial context inherent in the data (Yang et al., 2025). Conversely, training on entire tissue slices can introduce significant batch effects (Marx, 2021). To navigate this, we introduce a neighborhood-centric batching strategy. Our approach constructs each training mini-batch to represent a spatially contiguous local neighborhood. As detailed in Algorithm 2, we first compute a global nearest-neighbor graph for all spots in a slice. Then, for each mini-batch, the algorithm initiates with a random seed spot and iteratively incorporates its closest unassigned neighbors until the batch is full ($k$ spots). This ensures that each training step operates on a group of spatially proximal spots, creating an ideal setup for learning local tissue organization.

**Intrinsic Gene Reconstruction (IGR) Loss.** Our first objective, Intrinsic Gene Reconstruction, adapts masked language modeling (Devlin, 2018) to learn intra-spot gene co-expression patterns. We randomly mask 15% of tokens in each spot's gene sequence $T_i$ and train the model to reconstruct them

using the unmasked tokens from the same spot as context. The IGR loss is the negative log-likelihood over the set of masked tokens $M$:

$$\mathcal{L}_{IGR} = -\frac{1}{|M|}\sum_{j \in M} \log P(t_{i,j}|x_{i,L-1}), \tag{4}$$

where $t_{i,j}$ represents the $j$-th masked token in spot $i$.

**Contextual Gene Reconstruction (CGR) Loss** To explicitly model the spatial relationships between spots, Contextual Gene Reconstruction leverages the spatially coherent mini-batches. This task is guided by the biological principle that a spot's transcriptomic state is highly correlated with its immediate microenvironment (Long et al., 2023). Specifically, for a central seed spot $s_i$, we predict its masked gene tokens using only the aggregated information from its neighboring spots $N(s_i)$ as context. The final output embeddings $\{x_{i,L-1}^k | k \in N(s_i)\}$ from the neighboring spots are averaged to form a single neighborhood context vector, $h_i$. The model must then infer the identity of the masked tokens in the central spot from this external context. The loss is defined as:

$$\mathcal{L}_{CGR} = -\frac{1}{|M_i|}\sum_{j \in M_i} \log P(t_{i,j}|h_i), \quad h_i = \frac{1}{|N(s_i)|}\sum_{k \in N(s_i)} x_{i,L-1}^k, \tag{5}$$

By jointly optimizing both intrinsic and contextual reconstruction, the spatial-aware gene encoder learns to capture both the fundamental gene-gene interactions within a spot and the complex spatial dependencies that define tissue structure. The overall loss function can be formulated as:

$$\mathcal{L}_{Gene} = \mathcal{L}_{IGR} + \mathcal{L}_{CGR}. \tag{6}$$

## 3.3 HIERARCHICAL MULTI-SCALE CONTRASTIVE ALIGNMENT

Conventional contrastive learning methods often assume that most images possess distinct semantic content (Radford et al., 2021; Li et al., 2022), an assumption that holds in natural image domains. However, in WSIs, particularly across different scales within the same anatomical region, there exist complex inter-image correlations. Strictly isolating these images during training may lead to excessive fragmentation of the semantic space. To mitigate this issue, in addition to the standard **cross-modal contrastive loss** between patches and gene data, we further optimize a **cross-scale patch positioning loss** and an **intra-modal contrastive loss** between patches and regions (Figure 2b).

**Leveraging Paired Pathology and ST Data.** For paired data, we leveraged HEST (Jaume et al., 2024a), the largest pathological image and ST dataset. After filtering for human samples based on 10X Visium, we obtained 697K paired pathological images and gene expression data points.

**Pathological Vision Encoder.** The field of pathological vision foundation models has progressed rapidly, fueled by extensive large-scale training datasets (Chen et al., 2024b; Filiot et al., 2023; 2024; Xu et al., 2024b; Zimmermann et al., 2024; Saillard et al., 2024). However, the paired image data in HEST alone is insufficient to train a high-performing vision encoder from scratch. We selected UNI (ViT-L/16) (Chen et al., 2024b) as our visual backbone. Beyond its strong performance, using a pure vision encoder allows us to avoid the semantic bias inherent in image-text foundation models and the global-aggregation bias of bulk-RNA models. This ensures that our framework learns fine-grained spatial representations strictly driven by spatially-resolved molecular supervision, rather than inheriting priors from other modalities. (See Appendix A.3.4 for a detailed discussion).

**Cross-scale Patch Positioning (CSP).** Pathologists frequently zoom in and out during diagnostic workflows (Tran et al., 2025), motivating us to adopt cross-scale patch positioning as a pretext task to simulate their localization behavior and to establish connections between patch- and region-level representations. As shown in Figure 2b, to enable a shared vision encoder to process both patch and region images, we introduce a "pretext token" that allows the model to handle both input types in a unified manner. Given a patch, we treat it as a randomly selected sub-grid within a $3 \times 3$ grid layout. Based on its position, we crop a larger region and then resize it to $224 \times 224$ pixels.

In addition, the original patch is duplicated and resized to $16 \times 16$ pixels—matching the mini-patch size used in UNI. This process yields a tuple $\{P, ref, R, Pos\}$, where $P$ and $ref$ denote the original and resized patch, $R$ is the resized region, and $Pos$ represents the position label of the patch within the grid. When processing the region image, the embedded $ref$ is used as a pretext token and concatenated with the other mini-patch tokens from the region. The combined sequence is then fed into the vision transformer backbone. The cross-scale patch positioning loss can be formulated as:

$$\mathcal{L}_{CSP} = CE\Big(\big\{Sim\left(\mathbf{Z}_{pool,j}, \mathbf{z}_{ref}\right)\big\}_{j=0}^{8}, Pos\Big), \tag{7}$$

$$\mathbf{z}_{ref} = MLP\left(\mathbf{t}_{ref}\right), \quad \mathbf{Z}_{pool} = MLP\left(\mathbf{T}_{pool}\right), \quad \mathbf{T}_{pool} = GridPool(\mathbf{T}_{region}). \tag{8}$$

Here, $\mathbf{T}_{\text{region}}$ denotes the mini-patch tokens of the resized region $R$, and $\mathbf{t}_{\text{ref}}$ represents the token sequence derived from the resized patch $ref$. $\text{CE}(\cdot, \cdot)$ denotes the cross-entropy loss, while $\text{Sim}(\cdot, \cdot)$ represents cosine similarity. $\text{GridPool}(\cdot)$ is a pooling operation that aggregates $\mathbf{T}_{\text{region}}$ into a $3 \times 3$ grid structure. The pretext token degenerates into a standard learnable parameter when processing patch images. This design preserves the generality and adaptability of the model architecture while retaining the spatial information learned while processing region images.

**Inter-Modal Alignment.** We align pathology images and gene profiles by projecting them into a shared embedding space using InfoNCE-style contrastive learning (Oord et al., 2018). This technique, widely adopted in multimodal domains such as image-text alignment, has been proven effective. Specifically, for a batch of $M$ paired patch image-gene expression samples $\{(\mathbf{p}_i, \mathbf{g}_i)\}_{i=1}^M$, where $\mathbf{p}_i$ and $\mathbf{g}_i$ denote the $i$-th patch image and gene embedding, the symmetric loss function is defined as:

$$\mathcal{L}_{P-S} = -\frac{1}{2M} \sum_{i=1}^M \log \frac{\exp(\tau \mathbf{p}_i^T \mathbf{g}_i)}{\sum_{j=1}^M \exp(\tau \mathbf{p}_i^T \mathbf{g}_j)} - \frac{1}{2M} \sum_{n=1}^M \log \frac{\exp(\tau \mathbf{g}_n^T \mathbf{p}_n)}{\sum_{m=1}^M \exp(\tau \mathbf{g}_n^T \mathbf{p}_m)}, \quad (9)$$

where $\tau$ is the temperature parameter. The first term represents image-to-gene loss, and the second represents gene-to-image loss. The loss $\mathcal{L}_{P-S}$ aims to pull paired embeddings closer and push unpaired ones apart. Similarly, to align region-level images with gene expression profiles, we adopt the same contrastive learning objective, denoted as $\mathcal{L}_{R-S}$.

**Intra-Modal Alignment.** Beyond conventional inter-modal alignment, we introduce a strategy that aligns patch-level with region-level image embeddings. This approach effectively expands the vision encoder's receptive field. Furthermore, treating other regions as negative samples helps mitigate the representation collapse commonly observed in BERT-based methods (Li et al., 2020).

$$\mathcal{L}_{P-R} = -\frac{1}{2M} \sum_{i=1}^M \log \frac{\exp(\tau \mathbf{p}_i^T \mathbf{r}_i)}{\sum_{j=1}^M \exp(\tau \mathbf{p}_i^T \mathbf{r}_j)}. \quad (10)$$

The symbols in the formula represent similar meanings as above. $\mathbf{r}_i$ denotes the embedding of the $i$-th region image. The overall alignment loss function can be summarized as follows:

$$\mathcal{L}_{Align} = \mathcal{L}_{CSP} + \mathcal{L}_{P-S} + \mathcal{L}_{R-S} + \mathcal{L}_{P-R}. \quad (11)$$

# 4 EXPERIMENTS AND RESULTS

## 4.1 EXPERIMENTAL SETTINGS

**Tasks and Datasets.** We evaluated STAMP on four downstream tasks using six datasets. For **linear probing and unsupervised clustering**, we utilized the **DLPFC** (Maynard et al., 2021) and **HBC** (Xu et al., 2024a) datasets. **Gene expression prediction** was benchmarked on the **PSC** (Andrews et al., 2024), **HHK** (Lake et al., 2023), and **HER2+** (Andersson et al., 2021) datasets. All ST datasets were generated using the 10X Visium platform, except for HER2+, which used the Spatial Transcriptomics platform. Finally, the **LUAD-mutation** dataset from TCGA was used for WSI-level gene mutation classification. The goal is to predict binary mutation status (positive/negative) for four clinically relevant genes: EGFR, KRAS, STK11, and TP53. See the appendices for details on testing datasets (Appendix A.2.4) and comparative methods (Appendix A.4).

**Implementation Details.** We initially pretrain the gene encoder on the SpaVis-6M using masked token prediction objective (Devlin, 2018), optimizing only $\mathcal{L}_{IGR}$ for one epoch with a batch size of 256. We then continue pretraining on a spatially annotated subset of SpaVis-6M using spatial-aware sampling and $\mathcal{L}_{Gene}$ for one epoch, with each batch consisting of 24 mini-batches of 9 samples (216 samples per batch). Alignment pretraining is performed for 30 epochs with a batch size of 256 and gradient accumulation steps of 2. All pretraining uses the AdamW optimizer, and the learning rate is set to $10^{-4}$. All training is carried out on four NVIDIA A800 GPUs. To prevent data leakage, no downstream task data is accessed during pretraining. For specific details regarding the pretraining and downstream evaluation experiments, please refer to Appendix A.3.1 and A.3.3, respectively.

## 4.2 EXPERIMENTS ON LINEAR PROBING AND UNSUPERVISED CLUSTERING

To assess the cross-modal representational ability of STAMP, we use the **DLPFC** and **HBC** datasets with fine-grained labels. DLPFC features subtle visual differences across brain regions, requiring gene expression for accurate classification, making it ideal for testing cross-modal integration. In

Table 1: **Results of Linear Probing and Unsupervised Clustering.** The average and standard deviation of balanced accuracy (Bal. Acc., ↑), F1 score (Wgt. F1, ↑), and unsupervised clustering metrics (ARI ↑, NMI ↑) are reported. $\mathcal{V}$, $\mathcal{L}$, and $\mathcal{G}$ denote models pretrained on vision, language, and gene expression data, respectively. Note that $\mathcal{G}$ models are included explicitly as baselines for the gene modality component (STAMP$_G$). † denotes pretraining only on SpaVis-6M without multimodal alignment. ‡ indicates that the model adopts vision-gene alignment as described by Chen et al. (2024a). Hop+scS denotes the concatenation of features extracted by Hoptimus0 and scGPT-Spatial. STAMP$_{G,V,F}$ refers to the gene, vision, and fused embeddings (concat) generated by STAMP.

| Dataset / Method | $\mathcal{V}$ | $\mathcal{L}$ | $\mathcal{G}$ | DLPFC Bal. Acc. | DLPFC Wgt. F1 | DLPFC ARI | DLPFC NMI | HBC Bal. Acc. | HBC Wgt. F1 | HBC ARI | HBC NMI |
|---|---|---|---|---|---|---|---|---|---|---|---|
| CLIP | ● | ● | ○ | $0.415_{\pm0.090}$ | $0.507_{\pm0.057}$ | $0.101_{\pm0.075}$ | $0.169_{\pm0.104}$ | $0.625_{\pm0.032}$ | $0.692_{\pm0.019}$ | 0.274 | 0.449 |
| PLIP | ● | ● | ○ | $0.429_{\pm0.095}$ | $0.521_{\pm0.064}$ | $0.128_{\pm0.084}$ | $0.224_{\pm0.105}$ | $0.733_{\pm0.034}$ | $0.788_{\pm0.019}$ | 0.364 | 0.549 |
| CONCH | ● | ● | ○ | $0.454_{\pm0.101}$ | $0.540_{\pm0.088}$ | $0.124_{\pm0.058}$ | $0.215_{\pm0.091}$ | $0.704_{\pm0.023}$ | $0.764_{\pm0.007}$ | 0.406 | 0.576 |
| CHIEF | ● | ○ | ○ | $0.454_{\pm0.087}$ | $0.548_{\pm0.066}$ | $0.132_{\pm0.072}$ | $0.243_{\pm0.092}$ | $0.751_{\pm0.040}$ | $0.813_{\pm0.014}$ | 0.401 | 0.602 |
| GPFM | ● | ○ | ○ | $0.547_{\pm0.106}$ | $0.637_{\pm0.081}$ | $0.150_{\pm0.064}$ | $0.267_{\pm0.074}$ | $0.834_{\pm0.019}$ | $0.870_{\pm0.014}$ | 0.457 | 0.643 |
| UNI | ● | ○ | ○ | $0.544_{\pm0.112}$ | $0.621_{\pm0.090}$ | $0.144_{\pm0.082}$ | $0.260_{\pm0.098}$ | $0.859_{\pm0.016}$ | $0.885_{\pm0.007}$ | 0.499 | 0.646 |
| UNI2 | ● | ○ | ○ | $0.541_{\pm0.113}$ | $0.630_{\pm0.080}$ | $0.147_{\pm0.088}$ | $0.257_{\pm0.102}$ | $0.834_{\pm0.012}$ | $0.868_{\pm0.010}$ | 0.479 | 0.637 |
| Virchow2 | ● | ○ | ○ | $0.565_{\pm0.105}$ | $0.645_{\pm0.073}$ | $0.141_{\pm0.083}$ | $0.249_{\pm0.097}$ | $0.855_{\pm0.017}$ | $0.882_{\pm0.018}$ | 0.398 | 0.591 |
| Hoptimus0 | ● | ○ | ○ | $0.568_{\pm0.106}$ | $0.651_{\pm0.083}$ | $0.147_{\pm0.064}$ | $0.280_{\pm0.082}$ | $0.816_{\pm0.023}$ | $0.863_{\pm0.016}$ | 0.458 | 0.647 |
| GigaPath | ● | ○ | ○ | $0.558_{\pm0.099}$ | $0.640_{\pm0.068}$ | $0.170_{\pm0.076}$ | $0.291_{\pm0.094}$ | $0.833_{\pm0.016}$ | $0.867_{\pm0.014}$ | 0.486 | 0.637 |
| scGPT | ○ | ○ | ● | $0.441_{\pm0.072}$ | $0.551_{\pm0.040}$ | $0.179_{\pm0.063}$ | $0.248_{\pm0.071}$ | $0.547_{\pm0.024}$ | $0.617_{\pm0.007}$ | 0.194 | 0.348 |
| scGPT-Spatial | ○ | ○ | ● | $0.558_{\pm0.047}$ | $0.661_{\pm0.032}$ | $0.215_{\pm0.065}$ | $0.313_{\pm0.059}$ | $0.610_{\pm0.017}$ | $0.709_{\pm0.013}$ | 0.208 | 0.338 |
| Nicheformer | ○ | ○ | ● | $0.449_{\pm0.032}$ | $0.553_{\pm0.042}$ | $0.130_{\pm0.033}$ | $0.207_{\pm0.035}$ | $0.481_{\pm0.025}$ | $0.605_{\pm0.017}$ | 0.136 | 0.269 |
| **STAMP$^{\dagger}_G$ (ours)** | ○ | ○ | ● | $0.571_{\pm0.033}$ | $0.680_{\pm0.029}$ | $0.233_{\pm0.047}$ | $0.301_{\pm0.033}$ | $0.588_{\pm0.024}$ | $0.675_{\pm0.022}$ | 0.210 | 0.356 |
| PLIP‡ | ● | ○ | ● | $0.476_{\pm0.068}$ | $0.571_{\pm0.058}$ | $0.174_{\pm0.068}$ | $0.320_{\pm0.081}$ | $0.806_{\pm0.015}$ | $0.850_{\pm0.008}$ | 0.436 | 0.618 |
| CONCH‡ | ● | ○ | ● | $0.481_{\pm0.071}$ | $0.577_{\pm0.049}$ | $0.176_{\pm0.071}$ | $0.331_{\pm0.080}$ | $0.811_{\pm0.029}$ | $0.865_{\pm0.016}$ | 0.445 | 0.608 |
| Hop+scS | ● | ○ | ● | $0.568_{\pm0.106}$ | $0.659_{\pm0.087}$ | $0.161_{\pm0.064}$ | $0.289_{\pm0.074}$ | $0.825_{\pm0.024}$ | $0.869_{\pm0.013}$ | 0.455 | 0.645 |
| mSTAR | ● | ● | ● | $0.540_{\pm0.111}$ | $0.621_{\pm0.095}$ | $0.159_{\pm0.078}$ | $0.289_{\pm0.088}$ | $0.869_{\pm0.006}$ | $0.872_{\pm0.012}$ | 0.505 | 0.647 |
| OmiCLIP | ● | ● | ● | $0.381_{\pm0.090}$ | $0.463_{\pm0.059}$ | $0.09_{\pm0.081}$ | $0.164_{\pm0.101}$ | $0.738_{\pm0.021}$ | $0.792_{\pm0.012}$ | 0.294 | 0.474 |
| **STAMP$_G$ (ours)** | ● | ○ | ● | $\underline{0.658}_{\pm0.031}$ | $\underline{0.738}_{\pm0.023}$ | $\mathbf{0.369}_{\pm0.059}$ | $\underline{0.492}_{\pm0.042}$ | $0.659_{\pm0.012}$ | $0.745_{\pm0.006}$ | 0.416 | 0.537 |
| **STAMP$_V$ (ours)** | ● | ○ | ● | $0.624_{\pm0.065}$ | $0.707_{\pm0.038}$ | $0.246_{\pm0.057}$ | $0.399_{\pm0.058}$ | $\underline{0.872}_{\pm0.014}$ | $\underline{0.895}_{\pm0.009}$ | $\underline{0.526}$ | $\underline{0.674}$ |
| **STAMP$_F$ (ours)** | ● | ○ | ● | $\mathbf{0.721}_{\pm0.048}$ | $\mathbf{0.791}_{\pm0.024}$ | $\underline{0.342}_{\pm0.064}$ | $\mathbf{0.502}_{\pm0.041}$ | $\mathbf{0.899}_{\pm0.017}$ | $\mathbf{0.920}_{\pm0.009}$ | $\mathbf{0.590}$ | $\mathbf{0.708}$ |

contrast, HBC has clear visual distinctions between tissue types, allowing us to evaluate whether molecular supervision affects visual feature learning. We follow standard self-supervised and spatial transcriptomics practices, using **linear probing** and **unsupervised clustering** for evaluation. In this evaluation, we differentiate between the features from our unimodal encoders and their fused combination: STAMP$_G$ refers to features from the aligned Gene encoder, STAMP$_V$ refers to features from the aligned Vision encoder, and STAMP$_F$ refers to their Fused (concatenated) features.

As shown in Table 1, STAMP consistently outperforms the second-best model across all datasets, demonstrating that molecular-level multimodal contrastive learning enhances representation in both modalities. Vision-language pretraining tailored for pathology (*e.g.*, CLIP *vs.* PLIP) boosted performance on HBC but offers limited improvement on DLPFC due to its subtle visual differences. In contrast, fine-tuning with gene expression supervision (PLIP *vs.* PLIP‡; CONCH *vs.* CONCH‡) significantly improves performance on both datasets, highlighting the advantage of molecular guidance. STAMP achieved the best results, benefiting from a stronger architecture and larger training corpus. Appendix A.5.1 provides a more detailed analysis. We also present the results of linear probing, t-SNE, and unsupervised clustering for the DLPFC dataset in Figure 5. STAMP outperforms the best unimodal vision and gene encoders across all evaluation settings.

## 4.3 EXPERIMENTS ON GENE EXPRESSION PREDICTION

The **PSC**, **HHK**, and **HER2+** datasets were used to evaluate the performance of STAMP on gene expression prediction. Notably, the HER2+ dataset, built on the ST platform, enables evaluation of STAMP's cross-platform generalization ability. Specifically, we select the top 5,000 highly variable genes from each dataset as prediction targets. We use two strategies for gene expression prediction: STAMP$^*_{Reg}$, which performs linear probing on visual features for regression, and STAMP$_{Con}$, which follows the query-reference approach from BLEEP (Xie et al., 2024) (detailed in Appendix A.3.3).

As shown in Table 2, previous regression-based methods usually focus on predicting a few hundred genes and struggle when scaling up due to increased parameter size and training instability. In contrast, contrastive learning and linear probing for regression approaches are more stable. STAMP$_{Con}$

Table 2: **Results of Gene Expression Prediction.** The average and standard deviation of the Mean Squared Error (MSE, ↓), as well as the average Pearson correlation coefficient (PCC), are reported for the top 100 highly variable genes (PCC-V, ↑) and highly expressed genes (PCC-E, ↑). * denotes training of the frozen vision encoder via linear probing for regression.

| | Dataset
Method | Train.
Param. | PSC | | | HHK | | | HER2+ | | |
|---|---|---|---|---|---|---|---|---|---|---|---|
| | | | MSE ↓ | PCC-V ↑ | PCC-E ↑ | MSE ↓ | PCC-V ↑ | PCC-E ↑ | MSE ↓ | PCC-V ↑ | PCC-E ↑ |
| Regression-based | STNet | 12.08M | $0.330_{\pm0.077}$ | $0.110_{\pm0.035}$ | $0.153_{\pm0.033}$ | $1.357_{\pm0.675}$ | $0.039_{\pm0.037}$ | $0.052_{\pm0.034}$ | $1.190_{\pm0.515}$ | $0.171_{\pm0.105}$ | $0.159_{\pm0.094}$ |
| | EGN | 146.02M | $0.345_{\pm0.063}$ | $0.094_{\pm0.041}$ | $0.140_{\pm0.028}$ | $1.321_{\pm0.792}$ | $0.051_{\pm0.044}$ | $0.064_{\pm0.043}$ | $1.112_{\pm0.500}$ | $0.143_{\pm0.115}$ | $0.128_{\pm0.107}$ |
| | His2ST | 93.07M | $0.343_{\pm0.070}$ | $0.006_{\pm0.002}$ | $0.007_{\pm0.002}$ | $1.402_{\pm0.712}$ | $0.014_{\pm0.032}$ | $0.029_{\pm0.034}$ | $1.084_{\pm0.043}$ | $0.072_{\pm0.092}$ | $0.066_{\pm0.087}$ |
| | TRIPLEX | 95.20M | $0.338_{\pm0.083}$ | $0.004_{\pm0.001}$ | $0.005_{\pm0.002}$ | $1.372_{\pm0.693}$ | $0.045_{\pm0.002}$ | $0.044_{\pm0.074}$ | $1.073_{\pm0.540}$ | $0.217_{\pm0.105}$ | $0.210_{\pm0.094}$ |
| | CONCH* | 3.62M | $0.333_{\pm0.084}$ | $0.137_{\pm0.039}$ | $0.173_{\pm0.037}$ | $1.424_{\pm0.663}$ | $0.050_{\pm0.027}$ | $0.052_{\pm0.025}$ | $1.040_{\pm0.475}$ | $0.209_{\pm0.090}$ | $0.194_{\pm0.081}$ |
| | CHIEF* | 6.21M | $0.335_{\pm0.083}$ | $0.146_{\pm0.034}$ | $0.187_{\pm0.022}$ | $1.369_{\pm0.675}$ | $0.101_{\pm0.054}$ | $0.099_{\pm0.051}$ | $0.959_{\pm0.443}$ | $0.228_{\pm0.098}$ | $0.213_{\pm0.088}$ |
| | GPFM* | 9.32M | $0.320_{\pm0.085}$ | $0.186_{\pm0.032}$ | $0.226_{\pm0.019}$ | $1.337_{\pm0.679}$ | $0.142_{\pm0.053}$ | $0.127_{\pm0.050}$ | $0.922_{\pm0.439}$ | $0.258_{\pm0.094}$ | $0.237_{\pm0.086}$ |
| | UNI* | 9.32M | $0.323_{\pm0.081}$ | $0.166_{\pm0.019}$ | $0.195_{\pm0.025}$ | $1.340_{\pm0.693}$ | $0.134_{\pm0.053}$ | $0.113_{\pm0.041}$ | $0.930_{\pm0.444}$ | $0.245_{\pm0.099}$ | $0.226_{\pm0.093}$ |
| | GigaPath* | 17.13M | $0.319_{\pm0.082}$ | $0.185_{\pm0.034}$ | $0.223_{\pm0.025}$ | $1.335_{\pm0.687}$ | $0.137_{\pm0.045}$ | $0.116_{\pm0.043}$ | $0.916_{\pm0.440}$ | $0.253_{\pm0.094}$ | $0.235_{\pm0.087}$ |
| | Hoptimus0* | 17.13M | $0.317_{\pm0.082}$ | $0.192_{\pm0.038}$ | $0.230_{\pm0.040}$ | $1.380_{\pm0.683}$ | $0.134_{\pm0.057}$ | $0.120_{\pm0.046}$ | $0.926_{\pm0.445}$ | $0.252_{\pm0.100}$ | $0.233_{\pm0.093}$ |
| | mSTAR* | 9.32M | $0.325_{\pm0.083}$ | $0.190_{\pm0.033}$ | $0.221_{\pm0.013}$ | $1.332_{\pm0.659}$ | $0.147_{\pm0.045}$ | $0.126_{\pm0.039}$ | $0.918_{\pm0.445}$ | $0.260_{\pm0.035}$ | $0.235_{\pm0.084}$ |
| | OmiCLIP* | 3.62M | $\underline{0.315}_{\pm0.082}$ | $0.184_{\pm0.037}$ | $0.220_{\pm0.019}$ | $1.305_{\pm0.700}$ | $0.116_{\pm0.063}$ | $0.121_{\pm0.058}$ | $1.110_{\pm0.480}$ | $0.180_{\pm0.085}$ | $0.160_{\pm0.076}$ |
| | **STAMP$^*_{Reg}$** | 9.32M | $0.319_{\pm0.079}$ | $\underline{0.204}_{\pm0.036}$ | $\underline{0.248}_{\pm0.018}$ | $\underline{1.286}_{\pm0.705}$ | $\underline{0.160}_{\pm0.062}$ | $\underline{0.151}_{\pm0.056}$ | $\underline{0.904}_{\pm0.445}$ | $\underline{0.267}_{\pm0.096}$ | $\underline{0.248}_{\pm0.087}$ |
| CL-based | mclSTExp | 130.46M | $0.351_{\pm0.082}$ | $0.082_{\pm0.061}$ | $0.102_{\pm0.066}$ | $1.344_{\pm0.544}$ | $0.087_{\pm0.052}$ | $0.076_{\pm0.033}$ | $0.969_{\pm0.501}$ | $0.225_{\pm0.094}$ | $0.192_{\pm0.091}$ |
| | BLEEP | 25.45M | $0.366_{\pm0.098}$ | $0.095_{\pm0.072}$ | $0.119_{\pm0.072}$ | $1.338_{\pm0.636}$ | $0.108_{\pm0.054}$ | $0.107_{\pm0.037}$ | $1.002_{\pm0.479}$ | $0.210_{\pm0.088}$ | $0.203_{\pm0.082}$ |
| | **STAMP$_{Con}$** | 1.44M | $\mathbf{0.301}_{\pm0.087}$ | $\mathbf{0.218}_{\pm0.037}$ | $\mathbf{0.278}_{\pm0.020}$ | $\mathbf{1.233}_{\pm0.671}$ | $\mathbf{0.193}_{\pm0.070}$ | $\mathbf{0.186}_{\pm0.054}$ | $\mathbf{0.870}_{\pm0.442}$ | $\mathbf{0.279}_{\pm0.099}$ | $\mathbf{0.266}_{\pm0.090}$ |

achieved the best results across all metrics, showing stronger representation ability in both modalities and retrieving more accurate gene profiles. STAMP$^*_{Reg}$ also demonstrated that incorporating gene expression as an additional supervision signal enhanced performance, outperforming several pathology-specific vision encoders with larger parameter counts and more extensive pretraining data.

We visualized the actual and predicted expression of gene CYP1A2 (Figure 6) and CYP3A4 (Figure 7) from the PSC dataset. These genes regulate liver detoxification and drug metabolism, making them critical for drug therapy and disease prevention. Additionally, we provide visualizations of the HHK dataset (Figure 8 and Figure 9). STAMP showed greater biological heterogeneity than other methods.

## 4.4 EXPERIMENTS ON WSI CLASSIFICATION

We evaluated the WSI-level performance of STAMP on the **LUAD-mutation** dataset. To manage the computational demands of large WSIs, we employed ABMIL (Ilse et al., 2018) and performed five-fold patient-level cross-validation. As shown in Figure 3, STAMP achieves highly competitive or state-of-the-art performance across the four sub-tasks. Notably, STAMP surpassed Hoptimus0 and GigaPath, which utilized larger vision encoders. TANGLE focuses on WSI-level pretraining, using UNI as the feature extractor and ABMIL for aggregation, aligning WSIs with bulk RNA data from TCGA cohorts. These results showed that pretraining with genomic atlas data as supervision outperformed other self-supervised methods for certain tasks. Moreover, spatial transcriptomics offered finer-grained supervision than bulk RNA, which led to further performance gains.

To further validate the generalizability of STAMP, we expanded its evaluation to three additional WSI-level tasks across five datasets. In these five evaluations, STAMP outperformed its vision backbone on four of them, which strongly demonstrates that molecular supervision provides a generalizable signal that enhances performance on diverse WSI classification challenges. See Appendix A.5.3 for details.

## 4.5 ABLATION STUDY

Our ablation study (Table 6) validates our spatial-aware pretraining. While $\mathcal{L}_{IGR}$ alone enabled STAMP$_G$ to outperform Nicheformer (Schaar et al., 2024) and scGPT (Cui et al., 2024), it fell short of scGPT-Spatial (Wang et al., 2025a). Adding $\mathcal{L}_{CGR}$ allowed STAMP$_G$ to match scGPT-Spatial's performance, confirming the significant benefits of modeling spatial context.

During the alignment stage, we conduct ablation studies on our vision encoder (STAMP$_V$) using the DLPFC dataset. Conversely, we perform ablations on STAMP$_{Con}$ using the PSC dataset, where both vision and molecular representations contribute to performance. When integrating UNI and STAMP$_G$ using vanilla contrastive learning (*i.e.*, $\mathcal{L}_{P-S}$), the vision encoder exhibited a substantial performance improvement on the DLPFC dataset. Moreover, on the PSC dataset, this integration significantly outperformed traditional contrastive learning approaches (*e.g.*, BLEEP). Incorporating $\mathcal{L}_{R-S}$ led to slight performance improvements on both datasets, which we attribute to an expanded receptive

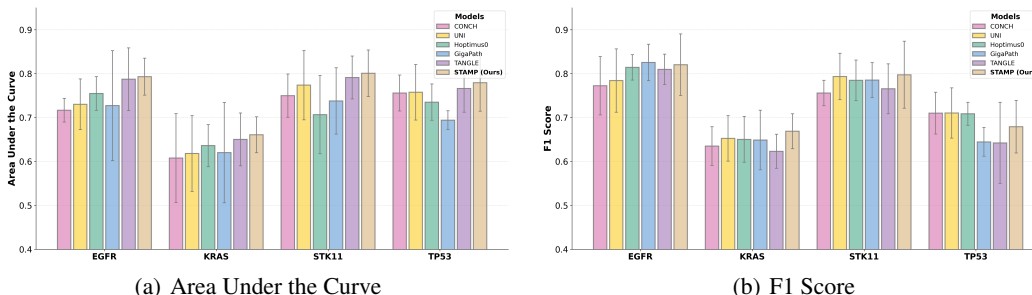

(a) Area Under the Curve  (b) F1 Score

Figure 3: **Results of MIL-based WSI Classification.** Comparison of STAMP and baselines for WSI-level gene mutation state classification using ABMIL on LUAD-mutation dataset.

field that mitigates representation collapse and enhances multi-scale spatial awareness. Moreover, the integration of $\mathcal{L}_{P-R}$ helped prevent representation collapse (Li et al., 2020) and improved the model's ability to learn strong features. $\mathcal{L}_{CSP}$ introduced a "zoom-in/zoom-out" spatial localization pretext task, enhancing the model's multi-scale spatial perception and enriching contrastive learning with more diverse negative samples, which resulted in consistent improvements across downstream tasks. In essence, this ablation study confirms that STAMP's unified alignment loss, synergistically combining objectives for spatial localization, inter-modal matching, and multi-scale consistency, effectively captures crucial spatial structures and molecular variations.

In addition, we performed comprehensive ablations on our two-stage training strategy and the SpaVis-6M dataset. We validated our design choices based on their impact on downstream performance (*i.e.*, Linear Probing and Unsupervised Clustering). The results show that the large-scale, multi-organ SpaVis-6M dataset and the Stage 1 pretraining are crucial for establishing a foundational understanding of gene co-expression, which is the basis upon which the second stage builds spatial context. The full results are deferred to Appendix A.6 and Table 7.

## 5 CONCLUSION AND DISCUSSION

**Conclusion.** We introduced STAMP, a spatial transcriptomics-augmented multimodal pathology framework that unifies pathology images and gene expression profiles via spatially-aware and multi-scale contrastive learning. By managing and utilizing the large-scale SpaVis-6M dataset, STAMP enables robust, task-agnostic representation learning and outperforms prior uni- and multimodal models across classification, clustering, and gene expression prediction tasks. These results demonstrate the value of integrating spatially-resolved molecular data for comprehensive computational pathology, highlighting STAMP's potential for broad clinical and research applications.

**Future Work.** These results highlight the potential of multimodal pretraining. Compared with other vision-language pretraining methods, the amount of data utilized in our study remains relatively limited, underscoring the need for large-scale data collection in future work. Specifically, such a collection need not be confined to cancer pathologies, which might unnecessarily constrain data availability. Instead, our findings suggest that non-cancerous data is indispensable for establishing robust gene expression baselines and learning generalized tissue representations. Furthermore, although we have demonstrated that models pretrained on 10X Visium data can be effectively transferred to the Spatial Transcriptomics platform, subsequent research should aim to develop more generalizable and robust models that encompass diverse sequencing technologies, platforms, and even other omics data.

**Limitations.** While STAMP delivers strong multimodal performance, it faces two primary limitations. First is data scale; the paired data, while the largest available, remains small compared to the hundreds of millions of pairs used in Vision-Language frameworks (*e.g.*, CLIP). This gap, driven by the high cost, privacy restrictions, and batch effects of ST assays, constrains the diversity of our training corpus and may limit STAMP's generalizability to rare tissues. Second is platform heterogeneity; STAMP is trained exclusively on Visium data. Although we show effective transfer to the Spatial Transcriptomics platform, the model was not exposed to other emerging technologies (*e.g.*, Xenium) and may underperform on their different capture chemistries, resolutions, and gene panels.

## 6 ETHICS STATEMENT

This research adheres to strict ethical and academic guidelines. All data used in this study, including for the construction of our SpaVis-6M dataset, were sourced exclusively from publicly available and fully de-identified repositories, as detailed in Appendix A.2.1. These resources are intended for non-clinical research purposes only, and their use, being an aggregation of pre-existing, anonymized public data, did not require institutional review board (IRB) approval. We aim for a positive societal impact by advancing computational pathology to support personalized medicine.

## 7 REPRODUCIBILITY STATEMENT

We are committed to ensuring the reproducibility of our work. The complete source code is provided in the supplementary materials, and upon acceptance, we will publicly release the pretrained model weights and the SpaVis-6M dataset. A comprehensive description of the SpaVis-6M dataset construction is available in Appendix A.2.2, while details on data processing for alignment and downstream tasks are provided in Appendices A.2.3 and A.2.4, respectively. Our methodology is thoroughly explained in Section 3, with key components like gene tokenization and spatial sampling presented as pseudocode in Algorithms 1 and 2. All experimental settings, including hyperparameters, computational resources, and implementation details, are documented in Section 4.1 and Appendix A.3.1. Finally, all baseline methods used for comparison are described in Appendix A.4 to facilitate fair and accurate replication of our results.

## 8 ACKNOWLEDGEMENTS

This project was funded by the National Natural Science Foundation of China 82090052.

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

# A   APPENDIX

## CONTENTS

## A.1   LLM USAGE DISCLOSURE

We used a Large Language Model (LLM) solely for minor text editing purposes, such as language polishing and typo correction. The LLM was not involved in research ideation, experimental design, analysis, or substantive writing.

## A.2   DATASETS

### A.2.1   ETHICAL CONSIDERATIONS

All SpaVis-6M resources are provided exclusively for non-clinical research purposes and must not be used to inform any diagnostic or therapeutic decisions. SpaVis-6M compiles only fully de-identified data drawn from open-access repositories (GEO, STimage-1K4M (Chen et al., 2024a), HEST-1K (Jaume et al., 2024a), SpatialOmics (Xu et al., 2024d), and STOmicsDB (Yuan et al., 2023)) under their respective data-sharing licenses. No direct or quasi-identifiers (*e.g.*, names, addresses, social security numbers) are included, and any attempt to re-identify individuals is strictly prohibited.

Because this work involves only aggregation and analysis of pre-existing, publicly available, de-identified datasets, no institutional review board submission or approval was sought or required. While the risk of re-identification is negligible, users must follow data-provider licenses and avoid attempts to recover sensitive information. We foresee no adverse societal impacts from the legitimate research use of SpaVis-6M.

### A.2.2 More Information for SpaVis-6M

We constructed SpaVis-6M, the largest Visium-based spatial transcriptomics dataset, to advance the training of robust spatially-aware gene encoders. SpaVis-6M (Figure 4a) aggregates 1,982 slices from 262 distinct sources, including GEO (1,008 slices), STimage-1K4M (Chen et al., 2024a) (309 slices), HEST-1K (Jaume et al., 2024a) (308 slices), SpatialOmics (Xu et al., 2024d) (302 slices), and STOmicsDB (Yuan et al., 2023) (55 slices). Spatial coordinates are available for 81.2% (1,611) of these slices. The dataset features a diverse organ distribution (Figure 4b), encompassing major organs such as the brain, skin, lung, kidney, pancreas, breast, and liver. SpaVis-6M spans a wide range of common tissues and patient conditions. We provide a detailed breakdown of this biological distribution (N=1,982 slices): **Cancerous** (855 slices, 43.1%), **Non-Cancerous** (1,127 slices, 56.9%), which includes **Diseased** (498, 25.1%), **Healthy** (539, 27.2%), and **Treated** (90, 4.5%) tissues. This balanced, heterogeneous collection provides a solid foundation for pretraining spatial-aware gene encoders. Spanning a wide range of common tissues and patient conditions (including healthy, diseased, and cancerous states), SpaVis-6M provides a solid foundation for pretraining spatial-aware gene encoders. Crucially, its scale enables the construction of a relatively unbiased gene expression baseline, offering a scientific basis for identifying "aberrantly expressed" gene patterns and characterizing whole-transcriptome alterations.

### A.2.3 Data for Alignment

The HEST dataset (Jaume et al., 2024a) was filtered to retain only the human data generated using the Visium platform. Additionally, only those spots located within tissues were kept. Spots with fewer than 100 detected gene expressions were removed as well. Since certain data from HEST will be used in downstream tasks, this data is excluded during the pretraining phase to prevent potential data leakage. For the pathology images, $224 \times 224$ pixel patches were extracted from the original WSIs based on the center point coordinates. This resulted in most patches covering a distance of 50-100 micrometers ($\mu m$), sufficient to encompass the corresponding spot (diameter of 55 $\mu m$). After these processing steps, 696,845 pairs of pathology images and spatial transcriptomic gene expression are available for alignment pretraining, sourced from 316 slices across 16 different tissues or organs. We provide a detailed breakdown of the biological distribution of these 316 slices: **Cancerous** (137 slices, 43.4%) and **Non-Cancerous** (179 slices, 56.6%), which includes **Diseased** (110, 34.8%), **Healthy** (43, 13.6%), and **Treated** (26, 8.2%) tissues. This biologically balanced cohort ensures the model learns diverse signals beyond just cancer pathology.

### A.2.4 More Information about Downstream Datasets

**DLPFC**: The human dorsolateral prefrontal cortex (DLPFC) dataset (Maynard et al., 2021) comprises 12 slices from three healthy donors. Each spot was categorized into seven classes: white matter (WM) and layers L1-L6, resulting in 47,329 data pairs. In the DLPFC dataset, except for the WM region, which shows distinct visual differences from other regions, the visual differences between the remaining regions are subtle, making differentiation more challenging.

**HBC**: The Human Breast Cancer (HBC) dataset (Xu et al., 2024a) comprises a single slice from an invasive ductal carcinoma. Each spot is annotated by experienced pathologists based on H&E staining, resulting in 20 distinct morphologically defined regions, which are further grouped into four major categories: ductal/lobular carcinoma in situ (DCIS/LCIS), invasive ductal carcinoma (IDC), tumor edge (regions with low malignant features), and healthy tissue. To increase the difficulty and granularity of our downstream analyses, we use the 20-region sublabels as the ground-truth labels for linear probing classification and unsupervised clustering.

**PSC**: The Human Primary Sclerosing Cholangitis (PSC) dataset (Andrews et al., 2024) consists of four tissue slices from a patient with primary sclerosing cholangitis, resulting in a total of 9,254 paired

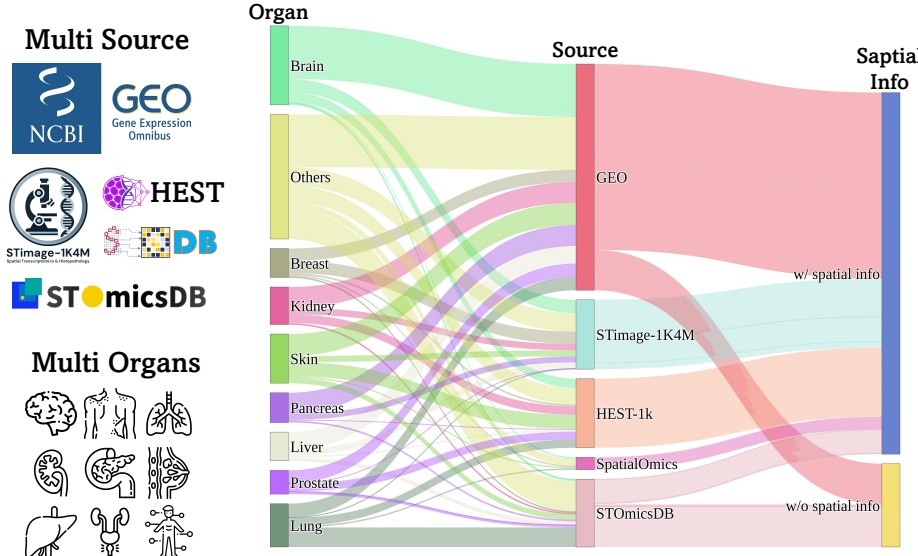

(a) Organ and Data Source Distribution across SpaVis-6M.

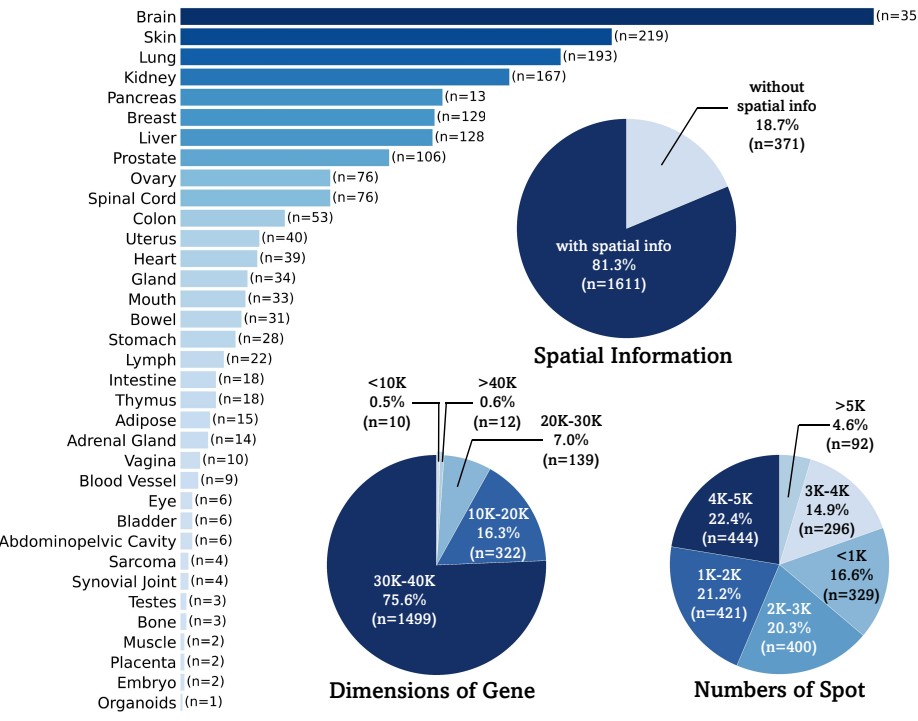

(b) Gene Dimension, Spot Number, and Spatial Information Statistics of SpaVis-6M.

Figure 4: Visium-integrated Spatial Transcriptomics Dataset (SpaVis-6M): Comprehensive Overview.

histological images and gene expression data. After the quality control, these four slices retained 2,377, 2,342, 2,275, and 2,260 pairs of histological images and gene expression, respectively.

**HHK**: The Human Healthy Kidney (HHK) dataset (Lake et al., 2023) consists of six tissue slices from six healthy kidney donors, resulting in a total of 15,401 paired histological images and gene expression data. After the quality control procedures, these six slices retained 1,033, 954, 2,622, 4,162, 3,623, and 3,007 pairs of histological images and gene expression, respectively.

**HER2+**: The HER2-positive breast tumor dataset (Andersson et al., 2021) (HER2+) consists of 36 slices from eight patients. Following ST-Net (He et al., 2020), we reserved 32 slides from seven patients, resulting in 11,509 data pairs. Unlike the previous datasets, HER2+ was measured using the Spatial Transcriptomics platform (Ståhl et al., 2016). Notably, during the pretraining phase, the model did not include any data based on Spatial Transcriptomics technology. The purpose of adding the HER2+ dataset is to assess the generalization capability of the STAMP across different sequencing technologies.

**LUAD-mutation:** The LUAD-mutation dataset consists of 692 Fresh Frozen WSIs from 437 patients in TCGA-LUAD. Following DeepPATH (Coudray et al., 2018), we aim to predict the WSI mutation state (positive/negative) in four specific genes: EGFR, KRAS, STK11, and TP53.

**UBC-OCEAN:** The UBC Ovarian Cancer subtypE clAssification and outlier detectioN (UBC-OCEAN) dataset (Asadi-Aghbolaghi et al., 2023) comprises 538 ovarian carcinoma WSIs. Each slide is classified into one of five histological subtypes: clear cell carcinoma (CC, 99 WSIs), endometrioid carcinoma (EC, 124 WSIs), high-grade serous carcinoma (HGSC, 221 WSIs), low-grade serous carcinoma (LGSC, 47 WSIs), and mucinous carcinoma (MC, 46 WSIs).

**TCGA-NSCLC:** The TCGA-NSCLC dataset contains 1,041 non-small cell lung cancer WSIs, including 530 lung adenocarcinoma (LUAD) and 511 lung squamous cell carcinoma (LUSC) slides. The task is to predict the histological subtype (LUAD vs. LUSC) from each WSI.

**PANDA:** The Prostate cANcer graDe Assessment (PANDA) dataset (Bulten et al., 2022) includes 10,616 prostate biopsy WSIs, each annotated with an International Society of Urological Pathology (ISUP) grade or labeled as normal, resulting in six classes.

## A.3   DETAILED EXPERIMENT SETTINGS

### A.3.1   EXPERIMENTAL SETTINGS FOR PRETRAINING

Pretraining for the gene encoder was conducted using four NVIDIA A800 GPUs (80 GB). The training of the gene encoder consists of two stages. In the first stage, we pretrain the gene encoder on the entire SpaVis-6M dataset by optimizing only the intra-spot masked gene token prediction loss ($\mathcal{L}_{IGR}$), thereby enhancing the model's general understanding of spatial transcriptomics data. In the second stage, building upon the pretrained weights from the first stage, we adopt a spatially-aware sampling strategy (Algorithm 2) to jointly optimize both intra-spot ($\mathcal{L}_{IGR}$) and inter-spot ($\mathcal{L}_{CGR}$) masked gene token prediction losses, enabling the model to learn spatial awareness. The configurations for this pretraining, including hyperparameters and setup details, are thoroughly outlined in Table 3.

### A.3.2   PRETRAINING FOR ALIGNMENT

All alignment pretraining experiments were conducted using four NVIDIA A800 GPUs (80 GB). Additional experimental configurations are detailed in Table 4. Furthermore, gene expression hidden states are extracted from the 12th transformer block, and mean pooling is applied along the sequence length dimension to obtain the encoded gene expression embeddings. The vision encoder follows the official feature extraction protocol.

### A.3.3   EXPERIMENTAL SETTINGS FOR DOWNSTREAM TASKS

We evaluated our STAMP on six downstream datasets and four tasks performed on a single NVIDIA A800 GPU (80 GB).

**Experiment Settings for Linear Probing** We employed different validation strategies based on dataset characteristics for the linear probing task. We performed leave-one-out cross-validation on the

---

**Algorithm 1:** Tokenization of Raw Gene Expression

**Input** : $\mathbf{mean} \in \mathbb{R}^{20310}$: per-gene average computed using only nonzero expressions across all data

$\mathbf{raw} \in \mathbb{R}^{B \times 20310}$: original gene expression (sparse-friendly)

$N \in \mathbb{Z}_+$: number of contextual tokens (e.g., 1500)

PAD_ID $= 0$: special padding token id

**Output** : $\mathbf{T} \in \mathbb{N}^{B \times N}$: tokenized gene indices

1 $\mathbf{raw} \leftarrow \text{ReplaceNaN}(\mathbf{raw}, 0)$ ;      // Placeholders only; zeros will be ignored

2 **for** $i \leftarrow 0$ **to** $B - 1$ **do**

3    $\mathcal{I} \leftarrow \{\, j \in \{1, \ldots, 20310\} \mid \mathbf{raw}[i, j] > 0 \,\}$ ;     // Indices of nonzero genes only

4    **if** $|\mathcal{I}| = 0$ **then**

5      $\mathbf{T}[i] \leftarrow [\text{PAD\_ID}] \times N$ ;      // No expressed genes; all PAD

6      **continue**

7    $\mathbf{v} \leftarrow \mathbf{raw}[i, \mathcal{I}]$ ;      // Values of nonzero genes

8    $c_i \leftarrow \sum \mathbf{v}$ ;      // Sum over nonzero only

9    $c_i \leftarrow c_i + (c_i == 0)$ ;      // Safety (should be nonzero if entered here)

10    $\mathbf{v} \leftarrow \mathbf{v} \times \frac{10000}{c_i}$ ;      // Normalize to 10,000 counts (nonzero only)

11    $\mathbf{v} \leftarrow \mathbf{v} \oslash \mathbf{mean}[\mathcal{I}]$ ;    // Mitigate batch effect; mean from nonzero samples

12    $\text{order} \leftarrow \text{argsort}(\mathbf{v}, \text{descending})$ ;      // Rank nonzero genes only

13    $\text{topk} \leftarrow \min(N, |\mathcal{I}|)$ $\text{sel} \leftarrow \mathcal{I}[\text{order}[: \text{topk}]]$ ;      // Top-topk gene indices

14    $\mathbf{T}[i] \leftarrow \text{sel}$ ;

15    **if** $len(\mathbf{T}[i]) < N$ **then**

16      $\mathbf{T}[i] \leftarrow \text{Concat}\big(\mathbf{T}[i], [\text{PAD\_ID}] \times (N - len(\mathbf{T}[i]))\big)$ ;      // PAD to length $N$

17    **end**

18 **end**

---

DLPFC dataset containing 12 slices and reported the mean and standard deviation across all folds. In contrast, the HBC dataset includes only one slice; therefore, we conducted five-fold cross-validation at the spot level, reporting the mean and standard deviation. We followed the standard linear probing protocol: after freezing the encoder, features were fed into a single-layer linear classifier. We used the embeddings before the projection layer for both models as input features. All models were trained with the cross-entropy loss and optimized using Adam. The learning rate was set to 1e-4 for the vision encoder, while a higher rate of 1e-3 was used for the gene encoder due to its slower convergence. Each model was trained for 100 epochs with an early stopping patience of 5 epochs.

**Experiment Settings for Unsupervised Clustering** For the unsupervised clustering task, we performed clustering independently on each of the 12 slices in the DLPFC dataset and reported the mean and standard deviation of the evaluation metrics. For the HBC dataset, which contains only a single slice, clustering was conducted solely on that slice; thus, no standard deviation is reported. All clustering analyses were performed using the Leiden algorithm as implemented in the scanpy library, with the resolution parameter uniformly set to 0.5.

**Experiment Settings for Gene Expression Prediction**

**Data Preprocessing and Evaluation Metric**: To account for heterogeneity across datasets in this study, we performed preprocessing separately for each dataset. Specifically, for each dataset, total-count normalization was applied to scale the transcript counts of each cell to 10,000, followed by a logarithmic transformation to compress the dynamic range of expression values. After normalization, all slices were merged, retaining only the intersection of gene sets present across slices. Based on this unified, normalized data matrix, we identified the top 5,000 highly variable genes using the Seurat v3 method, which selects genes exhibiting the greatest variability across cells to mitigate the curse of dimensionality and focus on informative features. Finally, for each original dataset, we extracted the expression matrix corresponding to these HVGs as the basis for downstream analyses, ensuring consistency of the feature space and reducing the impact of batch effects on subsequent results. Following the methodology of BLEEP (Xie et al., 2024), we report the Pearson Correlation

---

**Algorithm 2:** Spatial-aware Mini-batch Sampling

---

**Input** : $\mathcal{A}$: AnnData object with `obs['slide']`, `obs['position_row']`,
`obs['position_col']`
$k \in \mathbb{Z}_{>0}$: desired mini-batch size
$D_{\max} \in \mathbb{R}_{\geq 0}$: maximum allowed distance within a batch

**Output** : $\mathcal{B}$: list of filtered mini-batches

---

1   $\mathcal{B} \leftarrow []$ ;                               `// Initialize global batch list`

2   **foreach** *slide s in unique(*$\mathcal{A}$.obs*[ 'slide' ])* **do**

3      $\mathcal{I} \leftarrow \{i : \mathcal{A}.\text{obs}['\text{slide}']_i = s\}$ ;                   `// Indices on slide s`

4      $\mathbf{X} \leftarrow \{(r_i, c_i) \mid i \in \mathcal{I}\}$ from `obs['position_row']`,`obs['position_col']`;

5      Fit sklearn.neighbors.NearestNeighbors model on $\mathbf{X}$;

6      $U \leftarrow \mathcal{I}$ ;                            `// Unassigned spot indices`

7      **while** $|U| \geq k$ **do**

8          Pick and remove a hypothetical seed $u$ from $U$;

9          $G \leftarrow [u]$;

10         $N \leftarrow$ nearest neighbors of $u$ (sorted by distance);

11         **for** $v \in N$ **and** $|G| < k$ **do**

12             **if** $v \in U$ **then**

13                Append $v$ to $G$; remove $v$ from $U$;

14            **end**

15         **end**

16         Compute pairwise distances in $G$;

17         Move the true seed spot to the front of $G$ ;     `// Reset the center point to seed`
                        `spot`

18         Append $G$ to local batch list;

19      **end**

20      Remove any batch with max pairwise distance $> D_{\max}$;

21      Append valid batches to $\mathcal{B}$;

22   **end**

23   **return** $\mathcal{B}$

---

Table 3: Experiment Configurations for Gene Encoder Pretrain.

| | Hyperparameter | Value |
|---|---|---|
| **Model Architecture** | Vocab size | 20,310 |
| | Token dimensionality | 512 |
| | FFN dimensionality | 1024 |
| | Number of Transformer layers | 12 |
| | Max sequence length | 1,500 |
| | Number of attention heads | 16 |
| | Dropout | 0.0 |
| | Hidden act | ReLU |
| | LayerNorm eps | 1e-12 |
| **Phase One (intra loss)** | Optimizer | AdamW |
| | Scheduler | CosineWarmupScheduler |
| | Max learning rate | 1e-4 |
| | Min learning rate | 1e-5 |
| | Warm up steps | 5,000 |
| | Total Epochs | 1 |
| | Weight decay | 0.1 |
| | Global batch size | 256 samples |
| | Masking probability | 0.15 |
| **Phase Two (intra and inter loss)** | Optimizer | AdamW |
| | Learning rate | 1e-4 |
| | Total Epochs | 1 |
| | Weight decay | 0.1 |
| | Global batch size | 24 mini-batch / 216 samples |
| | Masking probability | 0.15 |

Table 4: Experiment Configurations for Alignment Pretrain.

| Hyperparameter | Values |
|---|---|
| Similarity function | Cosine similarity |
| Optimizer | AdamW |
| Scheduler | CosineWarmupScheduler |
| Max learning rate | 1e-4 |
| Min learning rate | 1e-5 |
| Warm up steps | 500 |
| Total epochs | 30 |
| Weight decay | 1e-3 |
| Globa batch size | 256 |
| Gradient Accumulation | 2 |
| Extraction layer | 12 |
| Pooling method | Mean |

Coefficient (PCC) for the top 100 highly variable genes (HVGs, PCC-V) and highly expressed genes (HEGs, PCC-E), as well as the Mean Squared Error (MSE) for the union of these HVGs and HEGs.

**Setting for $\text{STAMP}_{Con}$**: Using the retrial method proposed in BLEEP (Xie et al., 2024) for gene expression prediction is an effective approach to simultaneously evaluate the representational capacity of models for both pathology images and gene expression modalities. Our experiments adopted a strategy similar to CLIP-Adapter (Gao et al., 2024) to fine-tune our model on downstream datasets. Specifically, our adapter consists of two linear layers: the first layer halves the feature dimension, and the second layer restores it to the original size. When fine-tuning on downstream datasets, only the adapter layers and the projection layer are trainable, while all other parameters remain frozen, resulting in a total of 1.44 million trainable parameters. To simplify the application of STAMP to downstream tasks, we only continued to align the original patches and gene expression, without considering region patches. We employed a leave-one-out cross-validation method for data partitioning. The fine-tuned model is trained for 20 epochs, with early stopping applied using a patience of 5 epochs. The AdamW optimizer was used, with a learning rate set to 1e-3. The inference stage for predicting gene expression after fine-tuning strictly followed the query-reference strategy proposed by BLEEP (Xie et al., 2024).

**Query-Reference Strategy**: This approach predicts gene expression from pathology images using a multi-stage process. During inference, a fixed-weight (frozen) vision encoder converts input pathology images into query vectors $\mathbf{h} \in \mathbb{R}^{Q \times d}$. Concurrently, a pre-existing reference database, containing reference vectors $\mathbf{g} \in \mathbb{R}^{R \times d}$ generated by encoding the entire training set's gene expression data with a frozen gene encoder, is accessed. The cosine similarity metric quantifies the similarity between a given query vector and all reference vectors. Following this, the $K$ reference vectors most similar to the query are identified. A weighted method is then applied to the gene expression profiles associated with these top $K$ references to synthesize the predicted gene expression:

$$\hat{\mathbf{ep}}_q = \sum_{i \in K_q} w_i \mathbf{ep}_i, \quad w_i = \frac{\mathbf{h}_q \mathbf{g}_i^T}{\sum_{j \in K_q} \mathbf{h}_q \mathbf{g}_j^T}, \tag{12}$$

where $\hat{\mathbf{ep}}_q$ represents the predicted gene expression associated with the query image $q$, while $K_q$ denotes the set of the top $K$ nearest references for this query. Additionally, $\mathbf{ep}_i$ signifies the authentic gene expression linked to reference $i$.

**Setting for Linear Probing for Regression**: We adopt a linear regression approach analogous to linear probing in classification tasks to evaluate the performance of different vision encoders on the gene expression prediction task. All vision encoders are kept frozen during this evaluation. Since gene expression prediction is inherently more challenging than tissue type classification, we employ a multilayer perceptron consisting of three linear layers with ReLU activations for the regression task. All models are optimized using the mean squared error loss and the Adam optimizer, with a learning rate set to 1e-4. Leave-one-out cross-validation is used across all datasets to ensure robust evaluation. Each model is trained for 100 epochs, with early stopping applied using a patience of 3 epochs.

**Experiment Settings for Whole Slide Image Classification** For all WSI classification tasks, we used CLAM (Lu et al., 2021) to divide all WSIs into non-overlapping patches of $256 \times 256$ pixels at $20\times$ magnification. To meet the input requirements of the vision encoder, all patches were resized to $224 \times 224$ pixels. The simple yet effective ABMIL framework (Ilse et al., 2018) was utilized as the feature aggregation module, while the cross-entropy loss was employed to guide the training process. All models were set with a learning rate of 5e-4, used Adam as the optimizer, and were trained for 50 epochs with early stopping and a patience of 5. For the WSI Gene Mutation Classification task, which involved multiple WSIs per patient, we employed a more rigorous evaluation strategy. Specifically, five-fold cross-validation was performed at the patient level to prevent data leakage. Furthermore, when a patient had multiple WSIs, the patches from all their WSIs were stacked into a single, large bag. This strict patient-level, bag-merging method ensures no data leakage.

**Experiment Settings for Whole Slide Image Survival Prediction** We also used CLAM (Lu et al., 2021) to divide all WSIs into non-overlapping patches of $256 \times 256$ pixels at $20\times$ magnification. We utilized ABMIL (Ilse et al., 2018) as the feature aggregator and employed the Negative Log-Likelihood Loss (Zadeh & Schmid, 2020) for optimization. The model was trained for 30 epochs with a learning rate of 5e-5. Since each patient may have multiple WSIs, five-fold cross-validation was performed at the patient level to prevent data leakage.

### A.3.4 RATIONALE FOR VISION BACKBONE SELECTION

While recent multi-modal foundation models (*e.g.*, mSTAR Xu et al. (2024c), TITAN Ding et al. (2025), and MUSK Xiang et al. (2025)) have shown impressive results, we deliberately selected the uni-modal UNI encoder as our baseline based on three key scientific considerations:

**Mitigating Feature Space Misalignment**: Vision-language models align images with text, which primarily captures macro-level, linguistically describable phenotypes. In contrast, gene expression often correlates with subtle microenvironmental patterns that are hard to describe in natural language. Starting with an image-text model may introduce a "semantic bias," potentially filtering out critical visual features that are relevant to molecular prediction but are irrelevant to text descriptions. UNI provides an unbiased visual representation of how molecular signals can reshape.

**Addressing Granularity Mismatch**: Existing pathology-genomics models often rely on Bulk RNA-seq data, which averages gene expression across the entire slide. This global signal conflicts with our objective of capturing local spatial contexts. Using such models could bias STAMP toward global features, hindering its ability to learn fine-grained, spot-level associations.

**Ensuring Rigorous Comparison**: Crucially, UNI serves as the "visual anchor" for many SOTA models (including mSTAR and TITAN). By using the same root visual encoder, we ensure a fair comparison in which performance gains can be strictly attributed to our spatial transcriptomics-aware pretraining strategy, rather than to differences in the underlying visual architecture.

## A.4 COMPARISON METHODS

### A.4.1 COMPARISON METHODS IN LINEAR PROBING AND UNSUPERVISED CLUSTERING

Our comparative baselines include a range of state-of-the-art models and relevant approaches from both unimodal and multimodal domains:

**CLIP** (Radford et al., 2021): CLIP employs a dual-encoder architecture with separate vision and text encoders. The vision encoder is typically a Vision Transformer (ViT), with common variants including ViT-B/32, ViT-B/16, and ViT-L/14. Pretraining is conducted on 400 million image-text pairs collected from the internet. We use the ViT-B/32 version to be consistent with PLIP.

**PLIP** (Huang et al., 2023): PLIP is a pathogen-specific vision language model that undergoes continuous training, starting from the ViT-B/32 version of CLIP. It is pretrained on 208K pairs of pathological images and their corresponding texts collected from Twitter.

**CONCH** (Lu et al., 2024): CONCH is a vision-language model for computational pathology. It uses a ViT-B/16 backbone for the vision encoder and is pretrained on 1.17 million paired pathology image-text samples, enabling zero-shot and few-shot transfer in pathology tasks.

**PLIP[‡] and CONCH[‡]**: We followed the methodology proposed by Chen et al. (2024a), using spatial transcriptomics data as a supervisory signal. First, data from multiple independent samples were merged. The gene expression profiles were then normalized for library size and log-transformed to mitigate biases. Subsequently, 1500 highly variable genes were selected based on a predefined list to reduce dimensionality and focus on the most informative genes. Finally, the processed gene expression data were matched with their corresponding pathology image patches based on spatial location, creating paired inputs for the contrastive learning framework. Using this paired data, we independently fine-tuned the models on the DLPFC and HBC datasets, respectively. The hyperparameters used for this fine-tuning process were as follows: a learning rate of 5e-5, 15 training epochs, and a batch size of 256, with Adam used as the optimizer. After fine-tuning was complete, unified features were extracted from these models for the final validation.

**CHIEF** (Wang et al., 2024b): CHIEF is a general-purpose pathology foundation model built on 60,530 WSIs spanning 19 anatomical sites. It leverages two complementary pretraining strategies—unsupervised pretraining for tile-level feature extraction and weakly supervised pretraining for whole-slide pattern recognition. CHIEF learns transferable pathology representations useful for cancer detection, tumour origin identification, molecular profiling, and prognostic prediction.

**GPFM** (Ma et al., 2024): GPFM is a large-scale pathology foundation model trained on 190 million images derived from about 86,000 H&E whole slides spanning 34 major tissue types. It adopts a

unified knowledge distillation framework that combines expert distillation from multiple specialist models and self-distillation via local-global alignment. GPFM learns generalizable pathology representations designed to support diverse downstream clinical tasks.

**mSTAR** (Xu et al., 2024c): mSTAR is a multimodal pathology foundation model that integrates three modalities: whole-slide images, pathology reports, and gene expression data. It is trained on 26,169 slide-level multimodal pairs from 10,275 patients across 32 cancer types, amounting to over 116 million pathological image patches. mSTAR introduces a novel whole-slide pretraining paradigm, Multimodal Self-TAught PRetraining, which injects multimodal whole-slide context into patch-level representation learning, enabling comprehensive pathology feature extraction across diverse oncological tasks.

**OmiCLIP** (Chen et al., 2025a): OmiCLIP is a visual–omics foundation model designed to bridge the gap between hematoxylin and eosin (H&E) histology images and transcriptomics. It is trained on a curated dataset of 2.2 million paired tissue patches and transcriptomic profiles derived from Visium spatial transcriptomics data across 32 organs. OmiCLIP employs a novel cross-modal learning strategy where transcriptomic data are transformed into "gene sentences" by concatenating top-expressed gene symbols, effectively aligning histological features with genomic information.

**UNI** (Chen et al., 2024b): UNI is a vision-only foundation model for pathology, built on a ViT-L/16 architecture. It is pretrained on millions of whole slide images (WSIs) from diverse pathology datasets, aiming for strong generalization across tissue types and tasks.

**UNI2** (Chen et al., 2024b): UNI2 is an upgraded version of UNI, utilizing a larger ViT-H/14 backbone. It is pretrained on an even larger corpus of WSIs, further improving performance and data efficiency.

**Virchow2** (Zimmermann et al., 2024): Virchow2 adopts a ViT-H/14 architecture with 632 million parameters. It is pretrained using the DINO v2 self-supervised algorithm on over 3.1 million WSIs, leveraging a multi-view student-teacher strategy for robust feature learning in computational pathology.

**Hoptimus0** (Saillard et al., 2024): Hoptimus0 is a large ViT-based foundation model with the ViT-g/14 architecture. It is pretrained using the DINO v2 self-supervised algorithm on over 500K WSIs.

**GigaPath** (Xu et al., 2024b): GigaPath leverages a ViT-g/14 backbone. It is pretrained using the DINO v2 self-supervised algorithm on over 171K WSIs.

**scGPT** (Cui et al., 2024): scGPT is a generative pretrained transformer foundation model built on single-cell transcriptomic data. It is pretrained on a repository of over 33 million human cells across 51 organs and hundreds of studies, learning joint embeddings of genes and cells via self-supervised expression-prediction objectives. The model comprises roughly 53 million parameters, enabling efficient fine-tuning for diverse downstream tasks.

**scGPT-Spatial** (Wang et al., 2025a): scGPT-Spatial extends scGPT to spatial transcriptomics through continual pretraining on the newly curated SpatialHuman30M corpus, which contains 30 million spatially resolved profiles from Visium, Visium HD, Xenium, and MERFISH protocols. To handle protocol heterogeneity, it introduces a Mixture-of-Experts (MoE) decoder that routes samples through protocol-specific experts.

**Nicheformer** (Schaar et al., 2024): Nicheformer is a transformer-based foundation model that jointly ingests dissociated single-cell and spatial transcriptomics data to learn unified cellular representations. It is pretrained on the SpatialCorpus-110M dataset, comprising over 57 million dissociated cells and 53 million spatially resolved cells from 73 human and mouse tissues. By fine-tuning on spatial prediction tasks—such as spatial domain labeling and ecological-niche inference—Nicheformer demonstrates strong zero-shot and supervised performance, outperforming conventional spatial-omics pipelines.

### A.4.2   COMPARISON METHODS IN GENE EXPRESSION PREDICTION

Our comparative baselines include a diverse set of state-of-the-art models and methods, covering both regression-based and contrastive learning approaches:

**STNet** (He et al., 2020): STNet, a deep learning framework, is engineered for breast cancer analysis to predict gene expression by merging spatial transcriptomics data with pathology images. Input to the model consists of $224 \times 224$ pixel hematoxylin and eosin (H&E)-stained tissue patches, each correlating to a tissue spot of approximately 100 $\mu$m in diameter. DenseNet-121 handles image feature extraction, and these features subsequently feed into a fully connected layer to estimate the expression levels of 250 target genes. Our singular alteration was to this fully connected layer, enabling it to output the top 5000 HVGs.

**EGN** (Yang et al., 2023): EGN predicts spatial gene expression from tissue images by using similar examples called exemplars. It employs an extractor to find these exemplars, a Vision Transformer (ViT) to process image features, and specialized Exemplar Bridging blocks to integrate the exemplar information with the ViT representations for enhanced prediction accuracy.

**His2ST** (Zeng et al., 2022): His2ST leverages a dual-component architecture, combining Convolutional Neural Networks (CNNs) with Graph Convolutional Networks (GCNs), to predict spatial gene expression based on histopathological images. Initially, the CNNs are tasked with extracting localized features from the input images, thereby capturing the essential morphological characteristics of the tissue. Subsequently, the GCNs process these features to model the spatial relationships between adjacent regions, enabling the model to effectively discern and represent the spatial patterns of gene expression within the tissue environment.

**TRIPLEX** (Chung et al., 2024): TRIPLEX is a deep learning framework designed to predict spatial gene expression from Whole Slide Images by uniquely harnessing multi-resolution features. It captures cellular morphology from individual target spots, the local context from surrounding neighbor views, and the overall tissue organization from a global view of all spots. TRIPLEX employs separate encoders for each feature type. Then it integrates them using an effective fusion strategy, involving a fusion layer and a specialized fusion loss, to achieve accurate gene expression prediction.

**mclSTExp** (Min et al., 2024): The mclSTExp model utilizes a Transformer-based framework to specifically address the explicit modeling of spatial dependencies in spatial transcriptomics. Within this approach, individual spatial transcriptomics spots are conceptualized as 'words' forming a sequence, allowing self-attention mechanisms to integrate their positional and contextual information effectively. Furthermore, by integrating image-derived features through a contrastive learning strategy, mclSTExp enhances the precision of its spatial gene expression predictions, showing particular strength when characterizing intricate tissue structures.

**BLEEP** (Xie et al., 2024): The BLEEP framework leverages contrastive learning to forecast gene expression using pathology images. Central to its methodology, the model constructs a shared, compact latent representation derived from paired sets of pathology images and their corresponding gene expression profiles. When presented with a query image patch, BLEEP infers the associated gene expression by identifying its nearest neighbors within this learned embedding space, referencing a preestablished dataset. This approach enables precise and computationally efficient prediction of spatially resolved gene expression. Significantly, BLEEP surpasses existing methods in prediction accuracy and excels at preserving biological heterogeneity and demonstrating robustness against experimental artifacts.

### A.4.3 COMPARISON METHODS IN WHOLE SLIDE IMAGE CLASSIFICATION

We benchmarked several pathological vision encoders, namely **CONCH** (Lu et al., 2024), **CHIEF** (Wang et al., 2024b), **GPFM** (Ma et al., 2024), **UNI** (Chen et al., 2024b), **Hoptimus0** (Saillard et al., 2024), **GigaPath** (Xu et al., 2024b), **mSTAR** (Xu et al., 2024c), and **TANGLE** (Jaume et al., 2024b). Descriptions for these encoders, excluding TANGLE, are provided in Appendix A.4.1. TANGLE is a multimodal pretraining framework that learns whole-slide image (WSI) representations by aligning pathology slides with paired bulk RNA-seq data via contrastive learning. It employs UNI as the base vision encoder and utilizes ABMIL for feature aggregation. The resulting slide-level features are then aligned with bulk RNA expression profiles through a gene expression encoder, with both encoders trained jointly to project their respective modalities into a shared embedding space. TANGLE is pretrained on paired data from the TCGA dataset, spanning multiple tissue types with matched WSIs and transcriptomic profiles.

## A.5  MORE DETAILED ANALYSIS OF THE EXPERIMENTAL RESULTS

### A.5.1  MORE DETAILED ANALYSIS OF LINEAR PROBING AND UNSUPERVISED CLUSTERING

We benchmark STAMP against models from four categories: (1) Vision-Language pretraining models: **CLIP** (Radford et al., 2021), **PLIP** (Huang et al., 2023), and **CONCH** (Lu et al., 2024). (2) Vision-only pretraining models: **CHIEF** (Wang et al., 2024b), **GPFM** (Ma et al., 2024), **UNI** (Chen et al., 2024b), **UNI2** (Chen et al., 2024b), **Virchow2** (Zimmermann et al., 2024), **Hoptimus0** (Saillard et al., 2024), and **GigaPath** (Xu et al., 2024b). (3) Gene expression pretraining models: **Nicheformer** (Schaar et al., 2024), **scGPT** (Cui et al., 2024), and **scGPT-Spatial** (Wang et al., 2025a). (4) Vision-Gene pretraining models: This category includes **PLIP**[‡] and **CONCH**[‡] (variants finetuned on a single dataset following the approach of Chen et al. (2024a)); **Hop+scS**—a model that concatenates features from Hoptimus0 and scGPT-Spatial. (5) Vision-Language-Gene pretraining models: **mSTAR** (Xu et al., 2024c) and **OmiCLIP** (Chen et al., 2025a).

Reflecting the distinct characteristics of each dataset, gene expression-based methods demonstrate superior performance on the DLPFC dataset, whereas vision representation methods achieve stronger results on the HBC dataset. Unlike CLIP, both CONCH and PLIP are pretrained on an extensive corpus of pathology-specific image-text pairs. This large-scale, domain-specific pretraining yields substantial performance gains on the HBC dataset, with improvements reaching up to 17.28% in linear probing and 48.17% in unsupervised clustering. In contrast, the gains on the DLPFC dataset are more modest, at up to 9.40% for linear probing and 22.77% for unsupervised clustering. This performance disparity is attributed to the fact that the DLPFC dataset is characterized by subtle vision distinctions between brain regions, which limits the effectiveness of conventional vision-language pretraining approaches on this particular task. Adopting the methodology from (Chen et al., 2024a), we further pretrained PLIP and CONCH using spatial transcriptomics data as the supervisory signal. Experimental results demonstrate that this continued pretraining led to markedly enhanced performance for both PLIP and CONCH across two datasets. These toy experiments provided empirical support for developing STAMP.

Following its alignment pretraining, our model STAMP achieved state-of-the-art performance on both datasets. We denote its vision and gene encoder components as $STAMP_V$ and $STAMP_G$, respectively. Compared to the second-best performing vision model, $STAMP_V$ demonstrated improvements of up to 9.86% (linear probing) and 44.71% (unsupervised clustering) on the DLPFC dataset, and 1.51% (linear probing) and 8.23% (unsupervised clustering) on the HBC dataset. Similarly, $STAMP_G$ outperformed the next-best gene model; on the DLPFC dataset, improvements reached up to 9.28% (linear probing) and 58.37% (unsupervised clustering), while on the HBC dataset, gains were up to 8.03% (linear probing) and 98.10% (unsupervised clustering).

Furthermore, $STAMP_F$, which represents the concatenation of features from $STAMP_V$ and $STAMP_G$, surpassed its unimodal counterparts (*i.e.*, $STAMP_V$ and $STAMP_G$ individually) on seven out of a total of eight evaluation metrics across the two datasets.

Notably, when we concatenated features from the top-performing standalone unimodal models (Hoptimus0 and scGPT-Spatial) for downstream validation, the performance gains were minimal on the DLPFC dataset. This naive concatenation on the HBC dataset led to a performance decline compared to Hoptimus0 alone. This discovery indicates that the simple stacking of models inherently poses potential risks. Image features and gene expression profiles often reside in disparate latent spaces, and their direct concatenation can produce a highly heterogeneous feature distribution, likely causing the observed degradation in performance.

We also present the results of linear probing, t-SNE, and unsupervised clustering for the DLPFC dataset in Figure 5. STAMP outperforms the best unimodal vision and gene encoders across all evaluation settings. As illustrated in Figure 5a, we visualize the representations from the top-performing encoders of their respective modalities (namely, scGPT-Spatial for gene expression and Hoptimus0 for vision), alongside those from our fused model $STAMP_F$, and the ground truth. The performance of scGPT-Spatial surpasses that of Hoptimus0. Hoptimus0 can only differentiate white matter from other brain regions. In contrast, scGPT-Spatial can broadly distinguish between white matter, L1, L6, and L5, but exhibits poor discrimination for L4. $STAMP_F$, however, achieves markedly superior results compared to both these unimodal encoders, especially in effectively distinguishing the L4 region. In addition, we present t-SNE visualizations (Figure 5b) and unsupervised clustering

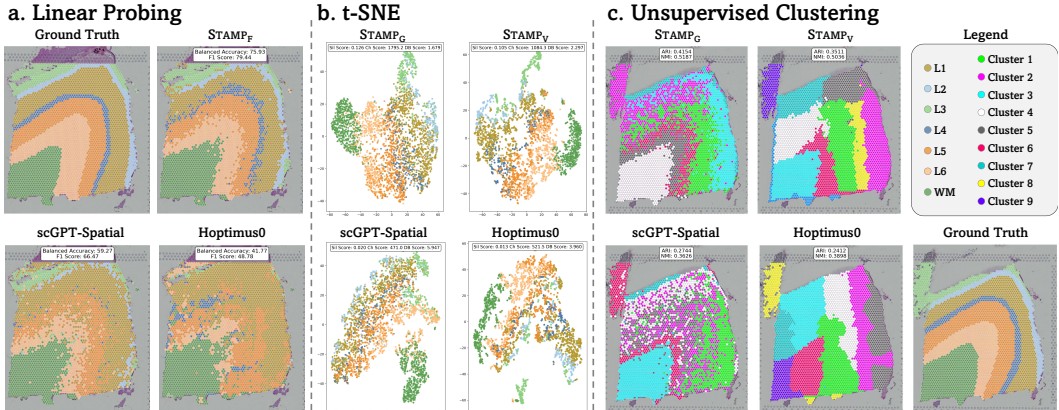

Figure 5: **Visualization of Linear Probing, t-SNE, and Unsupervised Clustering.** Results for STAMP, scGPT-Spatial, and Hoptimus0 on different samples: **a**. Linear probing on sample `151673`; **b**. t-SNE visualization on sample `151676`; **c**. Unsupervised clustering on sample `151675`.

results (Figure 5c) for two further samples. Invariably, STAMP demonstrated optimal performance across these additional examples.

### A.5.2 MORE DETAILED ANALYSIS OF GENE EXPRESSION PREDICTION

To assess STAMP's gene expression prediction capabilities, we conducted experiments on the PSC, HHK, and HER2+ datasets. Current regression-based methods, such as EGN, His2ST, and TRIPLEX, exhibit a substantial increase in parameters, surging to 146M, 93M, and 95M for EGN, His2ST, and TRIPLEX, respectively, when the number of target genes for prediction increases from a few hundred to 5000. These methods necessitate training from scratch, which, coupled with typically small datasets, results in suboptimal performance. STNet, despite its parameter count not increasing significantly due to an older vision encoder (DenseNet-121), also yields poor results because DenseNet-121 inadequately captures features from pathology images.

Conversely, our findings indicate that freezing pathology-specific vision encoders and applying simple linear probing for regression can lead to superior performance. Moreover, methods based on contrastive learning, like BLEEP and mclSTExp, also perform well, largely because their number of trainable parameters does not scale with the number of predicted genes. Motivated by these observations, we evaluate STAMP's proficiency in encoding bimodal data using two strategies: a linear probing for regression approach ($\text{STAMP}^*_{Reg}$) and a contrastive learning-based query-reference strategy ($\text{STAMP}_{Con}$). On all datasets, $\text{STAMP}_{Con}$ and $\text{STAMP}^*_{Reg}$ achieved the optimal and second-best results, respectively.

Figures 6&7&8&9 illustrate the visualization results for BLEEP, UNI, and our STAMP framework across a variety of genes and samples. Compared with BLEEP and UNI, $\text{STAMP}_{Con}$ and $\text{STAMP}^*_{Reg}$ not only demonstrate superior predictive performance but also better capture underlying biological heterogeneity.

### A.5.3 MORE EXPERIMENTS OF WSI-LEVEL TASKS

**Visualization of Attention Heatmaps**: To validate the biological interpretability of our model, we visualized attention weights for the EGFR, KRAS, STK11, and TP53 mutation prediction tasks on the LUAD-mutation dataset (Figure 10. Comparing STAMP with the baseline (UNI) highlights three key benefits. First, **Precise Localization**: STAMP accurately targets active tumor regions, whereas UNI is often distracted by useless areas like glass background and edges (*e.g.*, EGFR). Second, **Spatial Coherence**: STAMP generates smooth attention patterns that match the tissue structure, reducing the random noise seen in UNI (*e.g.*, KRAS). Third, **Biological Selectivity**: STAMP effectively ignores irrelevant parts like dead tissue, avoiding the problem where UNI highlights the entire image (*e.g.*, TP53). These results confirm that our method guides the model to learn truly meaningful features.

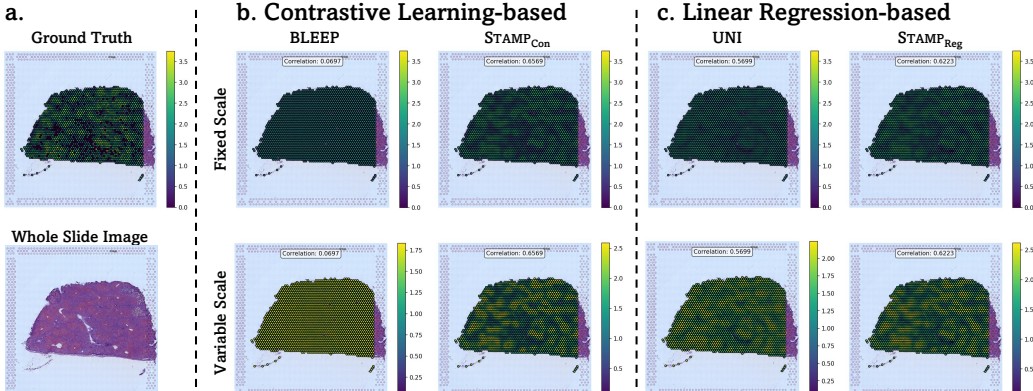

Figure 6: **Visualization of Gene Expression Prediction for the CYP1A2 Gene of sample** `C73_D1_VISIUM` **in PSC. a**. Ground truth and whole slide image. **b**. Predicted gene expression using contrastive learning-based methods. **c**. Predicted gene expression using linear regression-based methods. Each method is visualized with a fixed (top) and a variable (bottom) color scale.

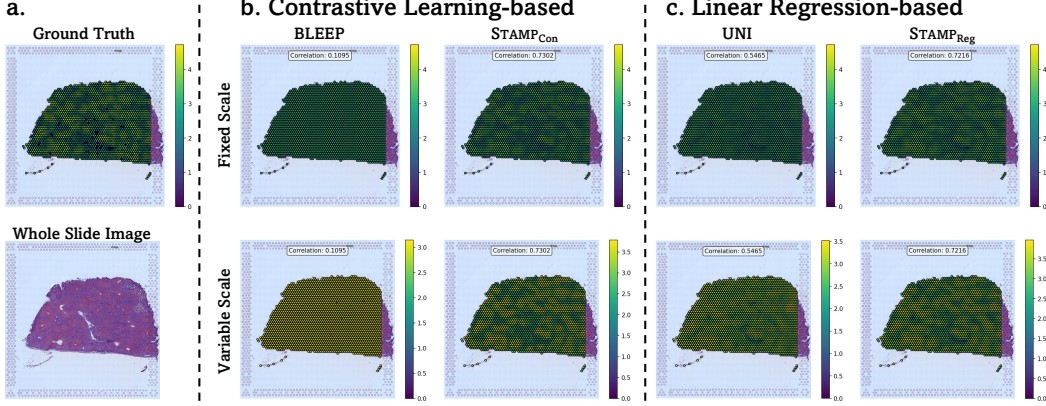

Figure 7: **Visualization of Gene Expression Prediction for the CYP3A4 Gene of sample** `C73_D1_VISIUM` **in PSC. a**. Ground truth and whole slide image. **b**. Predicted gene expression using contrastive learning-based methods. **c**. Predicted gene expression using linear regression-based methods. Each method is visualized with a fixed (top) and a variable (bottom) color scale.

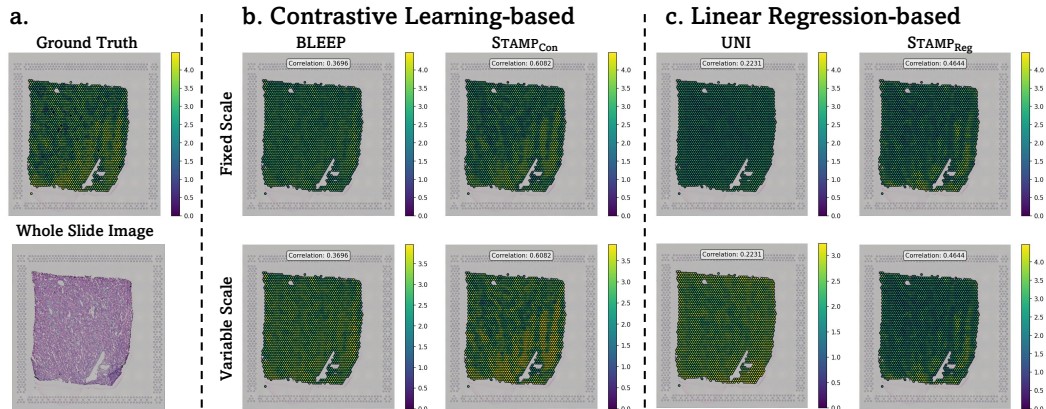

Figure 8: **Visualization of Gene Expression Prediction for the ATP1A1 Gene of sample `IU-F59` in HHK. a**. Ground truth and whole slide image. **b**. Predicted gene expression using contrastive learning-based methods. **c**. Predicted gene expression using linear regression-based methods. Each method is visualized with a fixed (top) and a variable (bottom) color scale.

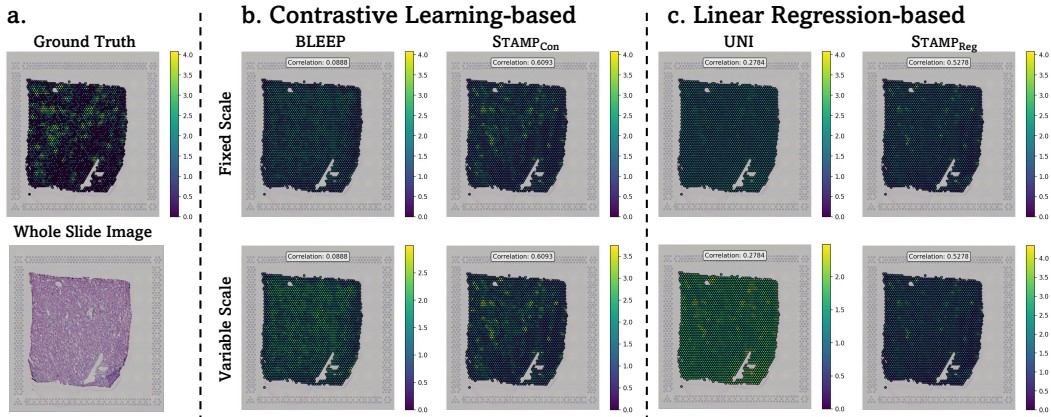

Figure 9: **Visualization of Gene Expression Prediction for the PODXL Gene of sample `IU-F59` in HHK. a**. Ground truth and whole slide image. **b**. Predicted gene expression using contrastive learning-based methods. **c**. Predicted gene expression using linear regression-based methods. Each method is visualized with a fixed (top) and a variable (bottom) color scale.

**Non-gene-centric Classification and Survival Prediction tasks**: We define the primary application boundary of our work as gene-centric WSI-level prediction tasks. Nevertheless, to comprehensively assess the generalizability of STAMP, we also present its performance on non-gene-centric classification and survival prediction tasks in Table 5. On these highly saturated benchmarks, STAMP achieves a performance uplift over its vision backbone (UNI) in four out of five tasks. This result indicates that our molecular supervision method not only excels in its target domain (gene-related tasks) but also learns representations that effectively generalize to broader computational pathology challenges. This demonstrates that STAMP is not a narrow specialist but rather a more powerful and versatile pathology foundation model.

## A.6    MORE DETAILED ABLATION STUDY

We conducted an ablation study to assess the contributions of our two-stage pretraining strategy and the large-scale SpaVis-6M dataset. The evaluation was performed across two datasets on two downstream tasks: Linear Probing and Unsupervised Clustering. Specifically, we compared our full approach against two ablated variants: (1) a model trained without the Stage 1 gene encoder

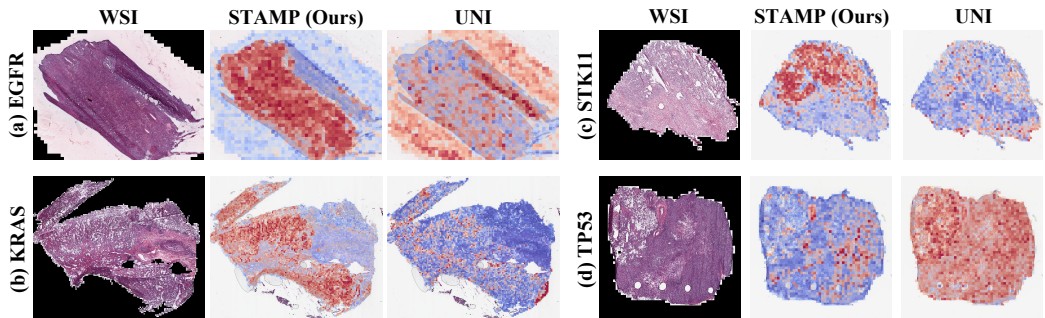

Figure 10: **Visualization of Attention Heatmaps on LUAD-mutation Dataset.** Comparison of attention distributions between STAMP and UNI across four sub-tasks: **(a)** EGFR, **(b)** KRAS, **(c)** STK11, and **(d)** TP53. Red indicates high attention regions. Compared to the baseline UNI, STAMP generates spatially coherent attention maps that precisely localize viable tumor regions while suppressing background noise and artifacts.

Table 5: WSI-level results. We report Macro-AUC and Accuracy on three datasets for WSI classification and C-Index on two datasets for WSI survival prediction.

| Method \ Dataset | WSI Classification | | | | | | WSI Survival Prediction | |
|---|---|---|---|---|---|---|---|---|
| | UBC-OCEAN | | TCGA-NSCLC | | PANDA | | TCGA-LUAD | TCGA-LUSC |
| | AUC | ACC | AUC | ACC | AUC | ACC | C-Index | C-Index |
| PLIP | 0.9453 | 0.6915 | 0.9395 | 0.8444 | 0.9045 | 0.5835 | 0.5906 | 0.5464 |
| CONCH | 0.9724 | 0.7881 | 0.9723 | 0.9126 | 0.9220 | 0.6528 | 0.6233 | 0.6045 |
| CHIEF | 0.9609 | 0.7751 | 0.9634 | 0.8972 | 0.9225 | 0.6425 | 0.6208 | 0.6092 |
| GPFM | **0.9788** | 0.8123 | 0.9704 | 0.9155 | 0.9521 | **0.7321** | 0.6467 | 0.6300 |
| mSTAR | 0.9764 | 0.8067 | **0.9730** | **0.9270** | 0.9468 | 0.7004 | 0.6329 | 0.6323 |
| GigaPath | 0.9771 | **0.8178** | 0.9684 | 0.9078 | **0.9525** | 0.7224 | **0.6544** | 0.6205 |
| UNI | 0.9732 | 0.7937 | 0.9695 | 0.9203 | 0.9455 | 0.7042 | 0.6312 | 0.6273 |
| **STAMP (ours)** | 0.9743 | 0.8141 | 0.9662 | 0.9087 | 0.9471 | 0.7087 | 0.6385 | **0.6321** |

Table 6: **Ablation Study of STAMP**. † denotes pretraining solely on SpaVis-6M without multimodal alignment, while # indicates pretraining with optimization only on $\mathcal{L}_{IGR}$.

| Model | Loss Function | | | | | | DLPFC | | PSC | |
|---|---|---|---|---|---|---|---|---|---|---|
| | $\mathcal{L}_{IGR}$ | $\mathcal{L}_{CGR}$ | $\mathcal{L}_{P-S}$ | $\mathcal{L}_{R-S}$ | $\mathcal{L}_{P-R}$ | $\mathcal{L}_{CSP}$ | ACC ↑ | ARI ↑ | MSE ↓ | PCC-E ↑ |
| $\text{STAMP}_G^{\#}$ | ✔ | ✗ | ✗ | ✗ | ✗ | ✗ | 0.553 | 0.204 | - | - |
| $\text{STAMP}_G^{\dagger}$ | ✔ | ✔ | ✗ | ✗ | ✗ | ✗ | **0.571** | **0.233** | - | - |
| UNI | ✗ | ✗ | ✗ | ✗ | ✗ | ✗ | 0.544 | 0.144 | - | - |
| | ✔ | ✔ | ✔ | ✗ | ✗ | ✗ | 0.592 | 0.193 | 0.332 | 0.199 |
| | ✔ | ✔ | ✔ | ✔ | ✗ | ✗ | 0.588 | 0.204 | 0.323 | 0.226 |
| | ✔ | ✔ | ✔ | ✔ | ✔ | ✗ | 0.613 | 0.229 | 0.310 | 0.266 |
| $\text{STAMP}_{V/Con}$ | ✔ | ✔ | ✔ | ✔ | ✔ | ✔ | **0.624** | **0.246** | **0.301** | **0.278** |

Table 7: Ablation Study of Linear Probing and Unsupervised Clustering. $\dagger$ denotes pretraining solely on the gene data without multimodal alignment.

| Dataset | DLPFC | | | | HBC | | | |
|---|---|---|---|---|---|---|---|---|
| Method | Bal. Acc. | Wgt. F1 | ARI | NMI | Bal. Acc. | Wgt. F1 | ARI | NMI |
| $\text{STAMP}_G^\dagger$ (w/o SpaVis-6M) | $0.497_{\pm 0.052}$ | $0.604_{\pm 0.049}$ | $0.188_{\pm 0.059}$ | $0.279_{\pm 0.021}$ | $0.544_{\pm 0.027}$ | $0.601_{\pm 0.017}$ | 0.186 | 0.312 |
| $\textbf{STAMP}_G^\dagger$ **(ours)** | $\mathbf{0.571}_{\pm 0.033}$ | $\mathbf{0.680}_{\pm 0.029}$ | $\mathbf{0.233}_{\pm 0.047}$ | $\mathbf{0.301}_{\pm 0.033}$ | $\mathbf{0.588}_{\pm 0.024}$ | $\mathbf{0.675}_{\pm 0.022}$ | **0.210** | **0.356** |
| $\text{STAMP}_G$ (w/o Stage 1) | $0.482_{\pm 0.037}$ | $0.582_{\pm 0.034}$ | $0.225_{\pm 0.048}$ | $0.320_{\pm 0.047}$ | $0.472_{\pm 0.017}$ | $0.558_{\pm 0.017}$ | 0.190 | 0.334 |
| $\text{STAMP}_G$ (w/o SpaVis-6M) | $0.566_{\pm 0.047}$ | $0.641_{\pm 0.034}$ | $0.221_{\pm 0.044}$ | $0.284_{\pm 0.018}$ | $0.602_{\pm 0.009}$ | $0.670_{\pm 0.010}$ | 0.342 | 0.498 |
| $\textbf{STAMP}_G$ **(ours)** | $\mathbf{0.658}_{\pm 0.031}$ | $\mathbf{0.738}_{\pm 0.023}$ | $\mathbf{0.369}_{\pm 0.059}$ | $\mathbf{0.492}_{\pm 0.042}$ | $\mathbf{0.659}_{\pm 0.012}$ | $\mathbf{0.745}_{\pm 0.006}$ | **0.416** | **0.537** |
| $\text{STAMP}_V$ (w/o Stage 1) | $0.564_{\pm 0.084}$ | $0.654_{\pm 0.056}$ | $0.187_{\pm 0.071}$ | $0.348_{\pm 0.078}$ | $0.860_{\pm 0.013}$ | $0.882_{\pm 0.014}$ | 0.408 | 0.595 |
| $\text{STAMP}_V$ (w/o SpaVis-6M) | $0.585_{\pm 0.072}$ | $0.673_{\pm 0.054}$ | $0.209_{\pm 0.066}$ | $0.363_{\pm 0.060}$ | $0.866_{\pm 0.016}$ | $0.889_{\pm 0.015}$ | 0.422 | 0.620 |
| $\textbf{STAMP}_V$ **(ours)** | $\mathbf{0.624}_{\pm 0.065}$ | $\mathbf{0.707}_{\pm 0.038}$ | $\mathbf{0.246}_{\pm 0.057}$ | $\mathbf{0.399}_{\pm 0.058}$ | $\mathbf{0.872}_{\pm 0.014}$ | $\mathbf{0.895}_{\pm 0.009}$ | **0.526** | **0.674** |
| $\text{STAMP}_F$ (w/o Stage 1) | $0.606_{\pm 0.077}$ | $0.694_{\pm 0.048}$ | $0.239_{\pm 0.060}$ | $0.383_{\pm 0.068}$ | $0.849_{\pm 0.014}$ | $0.889_{\pm 0.014}$ | 0.519 | 0.672 |
| $\text{STAMP}_F$ (w/o SpaVis-6M) | $0.631_{\pm 0.061}$ | $0.723_{\pm 0.042}$ | $0.255_{\pm 0.053}$ | $0.421_{\pm 0.047}$ | $0.871_{\pm 0.014}$ | $0.890_{\pm 0.008}$ | 0.541 | 0.682 |
| $\textbf{STAMP}_F$ **(ours)** | $\mathbf{0.721}_{\pm 0.048}$ | $\mathbf{0.791}_{\pm 0.024}$ | $\mathbf{0.342}_{\pm 0.064}$ | $\mathbf{0.502}_{\pm 0.041}$ | $\mathbf{0.899}_{\pm 0.017}$ | $\mathbf{0.920}_{\pm 0.009}$ | **0.590** | **0.708** |

pretraining (w/o Stage 1), and (2) a model where the gene encoder was pretrained only on the HEST dataset (w/o SpaVis-6M). The results are presented in Table 7.

The results in Table 7 clearly demonstrate the critical contributions of both our two-stage pretraining strategy and the large-scale SpaVis-6M dataset. When comparing our full model with the variant pretrained without SpaVis-6M (w/o SpaVis-6M), we observe a consistent and significant performance drop across all metrics. This is evident even in the pre-alignment gene encoder ($\text{STAMP}_G^\dagger$), confirming that the scale and diversity of SpaVis-6M are essential for building a robust gene encoder, which in turn provides a higher-quality supervisory signal during the multimodal alignment phase. Furthermore, the importance of our two-stage approach is validated by comparing it against the w/o Stage 1 variant. Removing the initial, general pretraining stage and only training with the spatially-aware objective leads to a severe degradation in performance for the vision ($\text{STAMP}_V$), gene ($\text{STAMP}_G$), and fused ($\text{STAMP}_F$) models. This confirms that the first stage is crucial for establishing a foundational understanding of gene co-expression, upon which the second stage effectively builds spatial context. In summary, these ablation results empirically prove that both the large-scale dataset and the carefully designed two-stage pipeline are indispensable components for achieving the final state-of-the-art performance of STAMP.

## A.7 LIMITATIONS AND WIDESPREAD SOCIAL IMPACT

### A.7.1 LIMITATIONS

While SpaVis-6M represents the largest 10X Visium-based spatial transcriptomics resource assembled to date, and our STAMP framework delivers strong multimodal performance, several limitations remain.

**Data Scale and Diversity**: Although HEST is the largest pathology image-spatial transcriptomic dataset, its size is still small relative to the hundreds of millions of image-text pairs used in leading multimodal contrastive frameworks (*e.g.*, CLIP, CONCH). This gap arises from the high per-sample cost of spatial transcriptomics assays, patient-privacy restrictions on clinical cohorts, and pronounced batch effects across sequencing platforms. As a result, the diversity and breadth of our training corpus remain constrained, which may limit STAMP's ability to generalize to rare tissues or unseen experimental conditions.

**Platform Heterogeneity**: We have shown that models pretrained on 10X Visium data transfer effectively to the original Spatial Transcriptomics platform. However, our pretraining set did not represent other emerging spatial-omics technologies (*e.g.*, MERFISH, Xenium, Slide-seq) owing to their current scarcity in public repositories. Without explicit exposure to these modalities, the model may underperform when faced with new capture chemistries, spot sizes, or gene panels.

**Future Directions**: To overcome these challenges, we highlight two complementary paths:

1. **Dataset Expansion**. Curate and integrate larger, more heterogeneous spatial transcriptomics collections—incorporating multiple platforms, species, disease states, and tissue types—to enrich molecular and morphological diversity.

2. **Robust Multimodal Design**. Develop architectures and training objectives that explicitly account for cross-platform batch effects, varying resolution scales, and modality-specific noise, thereby improving transferability across experimental settings and clinical workflows.

By pursuing these directions, future work can further strengthen spatially-aware multimodal pathology models' generality and clinical utility.

### A.7.2 WIDESPREAD SOCIAL IMPACT

Our work aims to bridge molecular-level insights with computational pathology, advancing diagnostic precision and personalized medicine. By integrating spatial transcriptomics with histopathology images, STAMP provides a richer representation that can improve understanding of tumor microenvironments and disease mechanisms, potentially guiding more effective cancer treatments.

However, spatial transcriptomics' high costs and technical complexity currently restrict access to well-funded research centers, limiting equitable benefit. We envision that scalable, cost-effective computational methods like STAMP can democratize molecular pathology, enabling broader adoption in clinical and research contexts, including underserved regions.

Moreover, computational pathology and spatial transcriptomics rapidly evolve but remain nascent and challenging. Our contribution fosters deeper integration between these disciplines, promoting synergistic advances that accelerate discovery.

In summary, while our study advances multimodal pathology representation learning, ongoing efforts are needed to expand data diversity, improve model robustness, and ensure equitable, ethical application of these technologies for societal benefit.

