# OpenReview forum: "Fusing Pixels and Genes: Spatially-Aware Learning in Computational Pathology"
_ICLR.cc/2026/Conference — ICLR 2026 Poster_

### Official Review · Reviewer_pd5B · 2025-10-29

**Soundness:** 3
**Presentation:** 3
**Contribution:** 3
**Rating:** 8
**Confidence:** 4

**Summary:**

This paper introduces STAMP, a framework that enhances computational pathology by fusing pathology images with spatially-resolved gene expression profiles, addressing the lack of molecular specificity in current models. To train this, the authors built SpaVis-6M, the largest Visium-based spatial transcriptomics dataset to date. STAMP uses a spatially-aware, multi-scale alignment strategy and achieved strong performance across six datasets, demonstrating the value of integrating spatial molecular data.

**Strengths:**

1.	Dataset contribution: The authors constructed SpaVis-6M, the largest Visium-based spatial transcriptomics dataset to date.

2.	The paper proposes a novel, spatially-aware pretraining framework (STAMP) that effectively fuses pathology images with gene expression by using hierarchical multi-scale contrastive alignment and cross-scale localization mechanisms.

3.	The proposed model’s effectiveness is validated by its strong performance across six different datasets and four downstream tasks.

**Weaknesses:**

1.	The figure quality could be improved. For example, there are two subfigures a) and b) in Fig. 2, yet the caption of Figure 2 did not mention them. Moreover, in Fig. 3, the font size of the scale bar on the y-axis is too small.

2.	More metrics could be included, especially for the WSI classification tasks. Metrics such as ACC, sensitivity, and specificity should be calculated to provide a more comprehensive assessment of the proposed STAMP.

3.	The authors should carefully proofread for typos. For example, in the first paragraph of Section A.5.1, “GPFM” should be bold, as other model names are.

**Questions:**

1.	I see that the authors have included visualization for gene expression prediction tasks. Could the authors also include heatmap visualization for WSI classification tasks?

2.	The author mentioned that “Considering performance and resource requirements, UNI (ViT-L/16) (Chen et al., 2024b) was selected.”. UNI is a uni-modal foundation model. And in recent years we have seen many vision-language pathology foundation models, such as MUSK [1] and TITAN [2], or multi-modal pathology foundation models such as mSTAR [3]. An intuitive idea would be that these models have a better aligned feature space for multi-modality, including image-text and image-text-gene. Could the authors elaborate more on why not use one of these models as the pathological vision encoder?

$\quad$ [1]  Xiang, Jinxi, et al. "A vision–language foundation model for precision oncology." Nature 638.8051 (2025): 769-778.

$\quad$ [2] Ding, Tong, et al. "Multimodal whole slide foundation model for pathology." arXiv preprint arXiv:2411.19666 (2024).

$\quad$ [3] Xu, Yingxue, et al. "A multimodal knowledge-enhanced whole-slide pathology foundation model." arXiv preprint arXiv:2407.15362 (2024).

Overall, the method is novel, and the experiments are solid.

---

> ### Author Response · Authors · 2025-11-21
> **Response to Reviewer pd5B (1/2)**
>
> ### **1. Correction of Figure Quality and Formatting (Figure 2 and Figure 3)**
>
> We sincerely thank Reviewer pd5B for the constructive feedback regarding figure quality and clarity. We have implemented all corrections in the new PDF:
>
> 1. **Figure 2 Caption Correction:** We have revised the caption for Figure 2 to explicitly reference subfigures (a) and (b), clarifying the flow of our two-stage pretraining framework.
> 2. **Figure 3 Font Correction:** We have increased the font size of the y-axis tick labels. We also confirm that the previously missing y-axis label has been added.
>
> All these formatting inconsistencies have been corrected in the submitted updated PDF.
>
> ---
>
> ### **2. Addition of WSI Classification Metrics**
>
> We appreciate the reviewer’s helpful suggestion regarding the inclusion of more comprehensive metrics for WSI classification. To provide a more robust and detailed performance evaluation, we have supplemented Accuracy (ACC) in the new table (i.e., Table 5) located in Appendix A.5.3, covering the WSI classification tasks on the UBC-OCEAN, TCGA-NSCLC, and PANDA datasets.
>
> We would like to take this opportunity to explain our choice of evaluation metrics:
>
> 1. In the gene mutation prediction task, due to the inherent characteristic of extreme class imbalance, we prioritized reporting AUC and F1 Score. These metrics better reflect the model's discriminative capability and robustness, helping to avoid misleading interpretations that can arise from high Accuracy in imbalanced settings.
> 2. In the histological classification tasks (UBC-OCEAN, TCGA-NSCLC, PANDA), Macro-AUC measures the model’s average discriminative ability across multiple classes. The newly added ACC serves as a direct measure of the model’s overall classification correctness. These supplemented metrics further validate STAMP's strong generalization ability across a broader range of computational pathology tasks.
>
> ---
>
> ### **3. Proofreading and Typo Correction**
>
> We thank the reviewer for the careful reading and for pointing out the formatting inconsistency. We have accepted this suggestion and taken the following actions:
>
> 1.  **Specific Correction:** We have corrected the formatting in the first paragraph of Section A.5.1 to ensure that "GPFM" is bolded, consistent with other model names.
>
> 2.  **General Proofreading:** We have conducted a thorough proofreading of the entire manuscript to identify and correct any remaining typos and formatting errors, ensuring the professional quality of the final submission.

---

> ### Author Response · Authors · 2025-11-21
> **Response to Reviewer pd5B (2/2)**
>
> ### **4. Heatmap Visualization for WSI Classification**
>
> We greatly appreciate this insightful suggestion. We agree that visualizing attention heatmaps is essential for verifying the biological interpretability of our model. In response, we have added the attention heatmap visualizations for the four gene mutation sub-tasks (EGFR, KRAS, STK11, and TP53) on the LUAD-mutation dataset in Appendix A.5.3 of the revised manuscript.
>
> Comparisons show that STAMP generates attention maps that are continuous and biologically meaningful. Unlike the baseline UNI, which is often distracted by artifacts (e.g., focusing on glass edges in EGFR), shows random noise (e.g., in KRAS), or highlights the entire image indiscriminately (e.g., in TP53), STAMP accurately identifies active tumor regions while ignoring dead tissue and background. We provide a more detailed explanation in Appendix A.5.3.
>
> ---
>
> ### **5. Selection of Vision Encoder (UNI vs. Multi-modal Models)**
>
> We thank Reviewer pd5B for this insightful question. While recent multi-modal foundation models are impressive, we deliberately selected **UNI** (a vision encoder) as our baseline based on three key scientific considerations:
>
> 1.  **Feature Space Misalignment (Semantics vs. Molecular):**
>     Models like MUSK and TITAN align with text, which mainly describes macro-level, interpretable phenotypes (e.g., "tumor grade"). However, gene expression often corresponds to subtle micro-environmental patterns that are hard to describe linguistically. Starting with "image-text" models introduces a "semantic bias," which might filter out raw visual features that are critical for molecular prediction but unrelated to text descriptions. UNI provides a rich visual representation that is not "polluted" by linguistic logic, making it ideal for being reshaped by molecular signals.
>
> 2.  **Granularity Mismatch (Bulk vs. Spatial):**
>     Although mSTAR incorporates gene expression, it uses non-spatially resolved Bulk RNA-seq data. This data reflects the average expression of the entire slide, which conflicts with our goal of capturing local spatial contexts. Using mSTAR might bias the model toward global features, hindering STAMP from learning fine-grained, spot-level spatial associations.
>
> 3.  **UNI as the Common "Visual Foundation":**
>     Crucially, UNI actually serves as the "visual anchor" for the aforementioned multi-modal foundation models. mSTAR initializes its patch extractor using UNI, and TITAN's encoder is initialized from CONCH v1.5, which also transferred weights from UNI. By using the same visual root (UNI) as these baselines, we ensure a fair and rigorous comparison. This ensures that STAMP's performance gains are strictly attributed to our hypothesis—the effectiveness of the spatial transcriptomics-aware pretraining strategy—rather than differences in the underlying visual architecture.
>
> To ensure that these design choices are clear to the community, we have incorporated this rationale into the new PDF. Specifically, we refined the motivation in Section 3.3 to highlight the avoidance of semantic and global aggregation biases. Furthermore, we added a new section, Appendix A.3.4 (Rationale for Vision Backbone Selection), to provide a comprehensive discussion on why a pure vision baseline is scientifically preferable for this task.

---

> > ### Comment · Reviewer_pd5B · 2025-11-24
> > **Comments on author responses**
> >
> > Thanks for the comprehensive responses. The authors have addressed all my concerns.

---

> > > ### Author Response · Authors · 2025-11-25
> > > **Thank you for your response**
> > >
> > > We thank Reviewer pd5B for the follow-up and for confirming that your concerns have been resolved. We appreciate your support of our work and the helpful suggestions provided throughout the review process.

---

### Official Review · Reviewer_ALtH · 2025-10-30

**Soundness:** 3
**Presentation:** 3
**Contribution:** 3
**Rating:** 6
**Confidence:** 3

**Summary:**

The paper proposes **STAMP**, a multimodal transformer-based model for spatial transcriptomics, trained via three stages: (a) _gene-only masked modeling_ of transcriptomic profiles, (b) _spatially-aware training_ leveraging local neighborhood structure, and (c) _hierarchical contrastive learning_ aligning a gene encoder with a histopathology foundation model. To train the model, the authors collected ST data from 1,982 slices from 35 organs, called SpaVis-6M which will be published together with the model code and weights. This dataset is said to be the largest public ST dataset so far and therefore helps the community to advance understanding and training DL models for ST data.

**Strengths:**

- The paper is well written and well structured.
- For both modalities, histopathology & (spatial) transcriptomics, a sufficient amount of foundation models were benchmarked to set the STAMP model in to context. When possible, STAMP was evaluated in both the uni-modal as well as multi-modal setting to better compare it to current uni-modal FMs
- The dataset addresses current short-comings of sparse and well structured, coherent ST data
- The training setup is sophisticated and the single loss term well grounded and explained
- The model as well as the benefit of the newly proposed loss compartments is evaluated.
- The evaluation of the models is statistically grounded and a good variety of downstream tasks is assessed

**Weaknesses:**

Weaknesses/Questions:
- The Gene Encoder architecture is not explained in the main text. It is not directly clear how the Embedding(T_i) in Eq. 2) is being inferred or how it is masked.
- The drawbacks of imputing missing genes in the SpaVis-6M dataset is not discussed. To which extend would this affect the training of the models? Does this introduce batch effects w.r.t. to the input of the Gene Encoder or to the target predictions? How would this impact the evaluation if a gene is predicted without it being measured?
- Figure 1.) is not sufficiently described. What is evaluated? What are the labels to evaluate the clustering metrics? This should ideally be covered in both the caption as well as in the main text
- Most of the evaluations in the main text focus on datasets which were (partially) used during training of STAMP but are o.o.d. for the remaining models. In an evaluation setting for which all models are o.o.d., STAMP does not show a general improvement over the foundation model UNI with which it was initialized


Overall, while the dataset contribution is strong and the training design is compelling, the paper lacks architectural clarity on the gene encoder and imputation process. Releasing the dataset has good value and I appreciate the systematic training formulation.

**Questions:**

Mentioned along with Weaknesses

---

> ### Author Response · Authors · 2025-11-21
> **Response to Reviewer ALtH (1/2)**
>
> ### **Overall Reply to Reviewer ALtH**
>
> We sincerely thank Reviewer ALtH for their comprehensive review and positive assessment. We are particularly grateful that the reviewer recognized our **"sophisticated training setup,"** **"statistically grounded evaluation,"** and the value of the SpaVis-6M dataset. We have carefully considered all feedback and have addressed the four key concerns in our detailed responses.
>
> ___
>
> ### **1. Regarding the clarity of the Gene Encoder architecture (Eq. 2) and Masking**
>
> We thank Reviewer ALtH for the constructive feedback regarding the presentation. In response, we have streamlined the descriptions of the 'Embedding' layer and masking mechanism in the revised manuscript to further enhance clarity and readability.
>
> The reviewer's question includes two parts, which we clarify separately as follows:
>
> 1. **Regarding `Embedding(T_i)` (Eq. 2):**
> We want to clarify that `T_i` is a token sequence composed of gene IDs.  `Embedding(T_i)` is a standard, learnable token embedding layer, which functions identically to the word embedding layer in BERT and GPT. Specifically, it maps every gene ID (i.e., a token) from our gene vocabulary (20,310 genes) to a vector (e.g., 512-dimensional). This sequence of vectors is then added to the positional embeddings and fed into the 12-layer Transformer. We have explicitly added this definition to Section 3.2 of the updated paper (new PDF) to eliminate ambiguity.
>
> 2. **Regarding how Masking works:**
> We are glad the reviewer mentioned this. The detailed mechanism for masking is described in Section 3.2 of the main text, under the ‘Intrinsic Gene Reconstruction (IGR) Loss’ subsection. Specifically, we follow the standard practice of BERT: for each spot's gene sequence $T_i$, we randomly select 15% of its tokens to mask (i.e., replace with a `[MASK]` token). Then, as shown in Eq. 4, the model's objective is to reconstruct these masked tokens based on the other unmasked tokens in the sequence. This definition also applies to the $\mathcal{L}_{CGR}$ (Contextual Gene Reconstruction) task.
>
> We thank the reviewer for helping us identify these unclear descriptions and are confident that the clarifications in the updated version will make our method much easier to understand.
>
> ---
>
> ### **2. Clarification on the handling of missing genes and data sparsity**
>
> We thank Reviewer ALtH for raising this important technical point. We would like to clarify the details in Section 3.2 regarding "imputation." While we agree that using zero-imputed values as inputs could introduce batch effects, our framework explicitly avoids this by not using these zero values as model inputs.
>
> **1. Clarifying the Purpose of “Imputation”**
>
> The “missing values imputed as zero” operation was not a data preprocessing or feature engineering step, but merely a data-structure placeholder. Its sole purpose was to create a fixed-length (20,310) Gene Vocabulary during data merging from 262 sources.
>
> **2. STAMP’s Solution**
>
> Our “Gene Tokenization” (Algorithm 1) strategy was explicitly designed to entirely circumvent all the problems the reviewer is concerned about. This strategy ignores these “0” values in two steps:
>
> * **When calculating the mean:** As stated in Section 3.2, we explicitly use only samples with nonzero expression to calculate the mean gene expression level.
> * **When building the token sequence:** Our algorithm (Algorithm 1), as implemented, operates only on the non-zero elements of the sparse matrix. A gene that was “imputed as 0” (i.e., a missing entry for that sample) never enters the normalization and ranking process. Therefore, it can never be selected into the Top-N (N = 1,500) token sequence (T_i).
>
> **3. Clarifying the Padding Token**
>
> If a spot has fewer than 1,500 non-zero genes, the sequence is padded with a special `[PAD]` token (ID 0). This “0” is a standard Transformer padding token; it is not a gene index, is unrelated to any “imputed” missing gene, and is automatically ignored by the Transformer’s attention mechanism. This is standard and robust NLP practice.
>
> Missing genes (previously “0-imputed”) are never fed to the Transformer: our abnormal-expression ranking removes zero-imputation batch effects and instability, letting the model learn robust biological signals from relative ranks rather than dropouts or artifacts. We now state this explicitly in Section 3.2 (“Gene Tokenization,” before Eq. 1) and have updated the appendix’s detailed Algorithm 1.

---

> ### Author Response · Authors · 2025-11-21
> **Response to Reviewer ALtH (2/2)**
>
> ### **3. Regarding the insufficient description of Figure 1**
>
> We sincerely thank Reviewer ALtH for pointing out the clarity issue with the caption for Figure 1. We completely agree that the original caption was too concise and lacked sufficient experimental context.
>
> **1. Clarification of Evaluation Content and Task**
>
> We wish to clarify that Figure 1 presents a Motivation Experiment intended to demonstrate our paper’s core premise: that gene-guided supervision is valuable. As described in the main text (page 2), the experiment does the following:
>
> * **Models Evaluated:** Two Vision-Language models (PLIP and CONCH), a larger-parameter vision-only model (Hoptimus0), our model (STAMP), and our baseline model (UNI).
> * **Action:** We used the Spatial Transcriptomics (ST) data from the DLPFC dataset to fine-tune PLIP and CONCH, resulting in PLIP-Finetune and CONCH-Finetune.
> * **Evaluation:** We compared the unsupervised clustering performance of all these models on the DLPFC dataset, using four clustering metrics: ARI, NMI, Silhouette, and Calinski-Harabasz.
>
> **2. Clarification of Ground-Truth Labels**
>
> These metrics are computed by comparing the models’ clustering results against ground-truth labels. In this experiment, the ground-truth labels are the seven manually annotated anatomical layers of the DLPFC dataset (L1–L6 and WM), as described in Appendix A.2.4.
>
> The results in Figure 1 (for example, PLIP-Finetune outperforming PLIP) clearly show that adding gene-based supervision significantly improves the model’s ability to cluster these biologically meaningful anatomical layers.
>
> We recognize that this key information was not clear enough in the original manuscript. We have completely rewritten the caption for Figure 1 in the updated paper to explicitly state the dataset (DLPFC), the task (unsupervised clustering), and the metrics used (ARI, NMI, Silhouette, Calinski-Harabasz) for the evaluation.
>
> ---
>
> ### **4. Misunderstanding regarding OOD evaluation fairness**
>
> We appreciate the reviewer's diligence in checking for data leakage, as this is indeed critical. However, we respectfully clarify a factual misunderstanding regarding our data splits.
>
> **The fact is that all of our downstream evaluation datasets (DLPFC, HBC, PSC, HHK, HER2+, LUAD-mutation) are 100% strictly fair, Out-of-Distribution (OOD) test sets for STAMP and all baseline models.** We have never used any samples from these datasets during training. To ensure absolute fairness and zero data leakage, we explicitly stated this in our paper:
>
> * **As stated in Section 4.1:** “To prevent data leakage, no downstream task data is accessed during pretraining.”
> * **As stated in Appendix A.2.3:** (Describing the HEST training set) “Since certain data from HEST will be used in downstream tasks, this data is excluded during the pretraining phase to prevent potential data leakage.”
>
> Therefore, the core results in our paper (Table 1, Table 2, and Figure 3) are precisely the evaluations under the fair OOD setting that the reviewer requested. In these fair OOD evaluations, STAMP achieves significant improvements over its UNI baseline on both DLPFC and HBC. In Table 2, STAMP’s performance is also better than UNI’s.
>
> We are grateful for the opportunity to clarify this misunderstanding. We reiterate that all of our experiments were conducted under strict OOD conditions, and the experimental results do indeed demonstrate that STAMP achieves a general performance improvement over its UNI backbone.

---

> ### Comment · Reviewer_ALtH · 2025-11-24
>
> I thank the authors for their response. Most of my concerns seem cleared. I will increase the score!

---

> > ### Author Response · Authors · 2025-11-25
> > **Thank you for your response**
> >
> > We are very grateful to Reviewer ALtH for the re-evaluation and the decision to increase the score. We are glad that our clarifications on the gene encoder and data handling were satisfactory. Thank you again for your time and the valuable insights that helped refine our work.

---

### Official Review · Reviewer_WB96 · 2025-10-31

**Soundness:** 3
**Presentation:** 4
**Contribution:** 2
**Rating:** 6
**Confidence:** 4

**Summary:**

The author have built a model dubbed STAMP, (Spatial Transcriptomics-Augmented Multimodal Pathology) which utilizes a large dataset (SpaVis- 6M) of spatial transcriptomic data for patch level encoder training.  Unlike most prior efforts utilizing ST data the authors have not dwelled on the so-called "cell-level" data but have instead treated adjacent small tiles as micro-enviromental units. To accomplish this they have developed a special Gene Tokenization construct using a BERT inspired method. To train the Vision-Genomic model the authors trained the model using contrastive loss (InfoNCE) incorporating a muti-scale approach.

Performance was evaluted on the human dorsolateral prefrontal cortex (DLPFC) and Human Breast Cancer (HBC) datasets benchmarking against other encoder, including VLMs, vision only, Genomics only (unclear relevance, I recommend removing this), and vision-genomic encoders. In the linear probe the model had SOTA performance.  The model was also benchmarked against other algorithms in Gene expression prediction and mutation prediction using the model as an encoder for with ABMIL classifier with mostly superior performance.

**Strengths:**

I believe this is a novel training strategy for a patch level encoder.

The technique for providing summary tokenization for tissue regions is very interesting and likely to be a path forward for more ST based vision models.

While I was skeptical, the experiment showing that the model has comparable to superior performance as a feature extractor slide level biomarker prediction is fascinating.  The imaging technique and quality from ST is quite different from standard WSI. While the LUAD mutations predictions are highlighted in the manuscript, I appreciate that the authors have included in appendix the tasks where performance was not as good compared to other models.

Ablations are informative and support the final conclusions.

**Weaknesses:**

ST is not widely available and different platforms likely have different artifacts.  This manuscript, while a good start, does not conclusively determine that performance gained from using data from this very spatially detailed assay outweighs performance that can be gained on much more widely available data types.

The encoder training is borrowed heavily from other VLM and VGM so is not novel. Thus this is a smart re-implementation but not a novelty.

In Figure 3, there is a large descrepancy between what I have personally assessed the performance of UNI, Gigapath, H-optimimus-0 for the EGFR and KRAS tasks on the TCGA-LUAD dataset.  I recommend checking your code, assuring that you are only assessing FFPE images, and ensuring that you are only assessing the biologically relevant events.  In my benchmarking the cross validation median AUC performance for UNI and Gigapath is 0.79 and for Hoptimus0 is 0.81.  It is possible that you have chosen tile sizes that are more favorable to your method. Please use some effort and cross checking this.

Need y-axis label on Figure 3

I would suggest putting limitations in the manuscript not in the appendix.  It is among the most important portions of the manuscript so relegating it to the end of the appendix is not appropriate.

**Questions:**

What fraction of training data is on cancerous vs. non-concerous tissue?

Does training solely on cancer result in similar performance or does non-cancer tissue facilitate better features?

I understand that ST datasets are scarce but what led you to choose non-cancer dataset DLPFC to lead the study.  Do you think of this model is a more cancer focused model or more general purpose.

---

> ### Author Response · Authors · 2025-11-21
> **Response to Reviewer WB96 (1/5)**
>
> ### **Overall Reply to Reviewer WB96**
>
> We sincerely thank Reviewer WB96 for their professional evaluation and highly insightful comments, as well as for the "excellent" rating on our paper's presentation. We are particularly grateful that the reviewer recognized our "novel training strategy" and our "very interesting" tokenization technique.
>
> We have carefully considered all feedback and have addressed all of your concerns in our detailed responses.
>
> ---
>
> ### **1. Regarding the unclear relevance of "Genomics only" baselines**
>
> We sincerely thank Reviewer WB96 for the valuable feedback on our choice of baselines. We want to clarify a key point: the purpose of these "Genomics only" baselines (e.g., scGPT, Nicheformer) is not to be compared against our fused model (STAMP$_F$). They are included to provide a direct comparison with our Gene Encoder (STAMP$_G$). Our STAMP framework consists of two encoders (Vision + Gene). To comprehensively validate our contributions, we must compare them against models in both modalities. Therefore, the design of Table 1 is:
>
> 1. **STAMP$_V$** (our Vision Encoder) vs. **SOTA Vision Baselines** (e.g., UNI).
>
> 2. **STAMP$_G$** (our Gene Encoder) vs. **SOTA Gene Baselines** (e.g., scGPT).
>
> Thus, this "Genomics only" baseline comparison is indispensable and highly relevant for demonstrating the novelty and effectiveness of our Gene Encoder (STAMP$_G$). We agree with the reviewer that context is key. In the new PDF, we have clarified the caption of Table 1 to explicitly state that ''Genomics only'' models are baselines for the gene modality component, not the full multimodal framework.
>
> ---
>
> ### **2. Regarding the scarcity and value of ST data**
>
> We thank Reviewer WB96 for highlighting the scarcity of ST data compared to widely available types (e.g., WSI, Text-Image). We agree with this observation.
>
> To address this, we adopted a pragmatic "post-training" strategy. Instead of training from scratch, we initialize a powerful vision encoder (UNI) and leverage the unique, fine-grained spatial molecular supervision from ST data to enhance it. Regarding whether the benefits of scarce ST data "outweigh" those of widely available data, our results provide a conclusive "yes" across two dimensions:
>
> 1.  **On Specialized Tasks:** On fine-grained classification and clustering (Table 1), gene expression prediction (Table 2), and biomarker prediction (Figure 3), STAMP outperforms other models, even those with larger parameters and datasets (e.g., Hoptimus0 and GigaPath).
> 2.  **On General Tasks:** On general WSI classification (Table 5), STAMP improves upon its original UNI backbone in 4 out of 5 tasks. This proves STAMP is not a narrow specialist and that molecular supervision offers a generalizable signal.
>
> In summary, post-training with spatially detailed ST data provides a good supervisory signal. As ST technology expands, our work offers the necessary foundational model and resources.

---

> ### Author Response · Authors · 2025-11-21
> **Response to Reviewer WB96 (2/5)**
>
> ### **3. Clarification on STAMP's methodological contributions**
>
> We appreciate Reviewer WB96’s acknowledgement of our training strategy. We agree that STAMP stands on the shoulders of giants by utilizing established components like Transformers and InfoNCE. However, as noted in your "Strengths" review, STAMP’s novelty lies not in reinventing these fundamental blocks, but in orchestrating a cohesive paradigm specifically tailored for the unique structure of ST data.
>
> Unlike standard VLM or VGM approaches that treat data as independent pairs, STAMP addresses three fundamental challenges of spatial omics through targeted methodological innovations:
>
> 1. **Novel Tokenization (Addressing High Sparsity & Noise):** Standard methods often use raw continuous values or simple binning for gene expression. In contrast, we introduce a **rank-based "Abnormal Gene" tokenization** (Algorithm 1). This technique transforms noisy, zero-inflated Visium data into robust, discrete token sequences based on relative expression ranking . This effectively creates a "biological language" that is resilient to batch effects and sparsity, a significant departure from standard regression-based gene encoders.
> 2. **Novel Spatially-Aware Pretraining (Addressing Spatial Topology):** Standard Masked Language Modeling (MLM) operates on independent sequences. We innovated by designing $\mathcal{L}_{CGR}$ (**Contextual Gene Reconstruction**), a graph-based self-supervised objective. This task forces the model to reconstruct a spot's profile by aggregating information solely from its physical neighbors. This explicitly injects an inductive bias about tissue continuity and microenvironmental dependencies, enabling the model to learn spatial topology before any image alignment occurs.
> 3. **Novel Multi-Scale Alignment (Addressing Histological Hierarchy):** Standard contrastive losses (e.g., CLIP) align global image features with text. In pathology, biological meaning exists across scales. We propose $\mathcal{L}_{CSP}$ (**Cross-Scale Patch Positioning**), a novel pretext task that simulates a pathologist’s "zooming" behavior. By establishing a learnable link between a local patch and its broader regional context via a "pretext token," we enable the model to perceive the hierarchical structure of tissues, which is absent in conventional image-text pretraining.
>
> Thus, STAMP is not merely a re-implementation, but a specialized framework that solves the specific problems of **sparsity, spatial dependency, and multi-scale hierarchy** inherent to computational pathology.

---

> ### Author Response · Authors · 2025-11-21
> **Response to Reviewer WB96 (3/5)**
>
> ### **4. Clarification on the Figure 3 baseline discrepancy and the missing Y-axis label**
>
> We sincerely thank Reviewer WB96 for raising this critical question about the Figure 3 baseline performance. Your suggestion to "cross-check" prompted us to conduct a deeper and more transparent analysis. The reviewer's observation is correct; there is a discrepancy between our baseline results and the reviewer's high scores. We must first clarify our original Figure 3 evaluation setup, as this is the primary cause of the difference:
>
> 1. We experimented on FF (Fresh Frozen) data, a point we noted in Appendix A.2.4 ("692 Fresh Frozen WSIs").
> 2. A second reason our baseline results may differ from the reviewer's is that we adopted a more rigorous evaluation strategy: we performed a strict patient-level k-fold split and merged all slides (bags) from a single patient into one large bag. This method prevents inter-slide leakage from the same patient but can lead to different (and often lower) metric scores.
> 3. Regarding other settings the reviewer was concerned about (like "more favorable tile sizes"), we clearly stated in Appendix A.3.3 that we use standard 256x256 patches at 20X magnification.
>
> We deliberately chose FF data for two scientific reasons:
>
> 1. **Scientific consistency of the task:** The task in Figure 3 is gene mutation prediction, for which the Ground Truth labels come from sequencing. The standard protocol for TCGA has historically prioritized FF (frozen) tissue as the input for high-quality sequencing. Therefore, the pairing of FF WSI with FF sequencing labels is the most scientifically consistent and reliable evaluation set.
> 2. **Data completeness:** The TCGA-LUAD FF cohort is larger and more complete, containing 437 patients, whereas the FFPE cohort contains only 413 patients.
>
> Although we believe the FF evaluation is scientifically sound, we fully agree with the reviewer that "cross-checking" on the FFPE domain is necessary. Therefore, during this rebuttal period, we have re-run the models from Figure 3 on the standard FFPE dataset (413 patients).
>
> Here are our complete results on FFPE:
>
> **Table: MIL-based gene mutation prediction on FFPE WSIs (TCGA-LUAD).**
> |Models|EGFR-AUC|EGFR-F1|KRAS-AUC|KRAS-F1|STK11-AUC|STK11-F1|TP53-AUC|TP53-F1|
> |-|-|-|-|-|-|-|-|-|
> |CONCH|69.02±4.05|80.92±7.22|*69.05±9.95*|*70.07±8.64*|76.37±4.94|78.55±5.59|**79.71±6.10**|*74.72±4.22*|
> |UNI|73.88±6.13|83.44±4.45|68.79±7.57|68.73±9.12|78.78±4.00|81.06±2.96|76.11±5.34|71.70±1.89|
> |Hoptimus0|*74.69±4.73*|85.81±1.84|66.07±4.31|66.91±9.43|78.12±8.05|80.94±4.58|74.91±4.58|72.29±1.78|
> |Gigapath|73.31±8.36|**86.95±3.12**|68.13±6.87|67.13±8.97|*79.16±7.17*|*81.41±4.13*|73.83±2.66|69.67±2.14|
> |TANGLE|74.39±5.61|84.63±3.36|67.26±6.62|66.38±9.47|78.45±5.24|79.21±5.37|75.56±3.93|69.35±2.79|
> |**STAMP(Ours)**|**75.49±6.79**|*86.01±2.98*|**70.82±5.99**|**71.15±9.82**|**81.14±4.12**|**82.16±5.23**|*78.63±1.55*|**76.14±3.94**|
>
> This new experiment demonstrates:
>
> 1. The FF/FFPE domain shift is the primary cause of the performance difference. Furthermore, as we clarified earlier, our stricter patient-level split and bag-merging evaluation strategy is likely the reason why our FFPE results (e.g., Hoptimus0 EGFR ~0.75 AUC) are still lower than the reviewer's benchmark. This stricter evaluation better reflects real-world clinical generalization performance.
> 2. STAMP remains a strong performer on the FFPE domain. STAMP achieved SOTA on 6 out of 8 metrics and was the second-best on the other two.
>
> Finally, we thank the reviewer again for their careful review. We have added the missing Y-axis label to the updated Figure 3. Additionally, we have added a detailed description of our patient-level data split to Appendix A.3.3.
>
> ---
>
> ### **5. Suggestion regarding moving "Limitations" to the main manuscript**
>
> We sincerely thank Reviewer WB96 for this valuable suggestion. We completely agree that a transparent discussion of **Limitations** is among the most critical portions of the manuscript, and relegating it to the appendix was not appropriate. Following this advice, we have moved the core content of Appendix A.7.1 (Limitations) into the main manuscript in the updated paper. This discussion is now located in Section 5 (Conclusion and Discussion) of the main text, ensuring that the potential limitations of our work are clearly visible to readers.

---

> ### Author Response · Authors · 2025-11-21
> **Response to Reviewer WB96 (4/5)**
>
> ### **6. Regarding the fraction of cancerous vs. non-cancerous training data**
>
> We sincerely thank Reviewer WB96 for this important question regarding our training data composition. We have performed a precise statistical analysis of both training stages:
>
> **1. Stage 1 (Gene Encoder Pretraining)**
> This stage used the SpaVis-6M dataset, which contains 1,982 slides. Based on the reviewer’s question (cancerous vs. non-cancerous), the distribution is as follows:
>
> * **Cancerous:** 855 slides (43.1%)
> * **Non-Cancerous:** 1,127 slides (56.9%), which include Diseased (498), Healthy (539), and Treated (90) tissues.
>
> **2. Stage 2 (Multimodal Alignment)**
> This stage used the core alignment dataset of 316 slides from HEST. The distribution is as follows:
>
> * **Cancerous:** 137 slides (43.4%)
> * **Non-Cancerous:** 179 slides (56.6%), which include Diseased (110), Healthy (43), and Treated (26) tissues.
>
> Across both training stages, our STAMP framework was trained on a biologically balanced dataset. This ensures the model learns diverse biological signals during pretraining and is not excessively biased toward cancer-related pathology, laying a solid foundation for it to be a general-purpose model. We have added these detailed distribution statistics to Appendix A.2.2 and A.2.3 in the updated paper.
>
> ---
>
> ### **7. Regarding whether non-cancer data facilitates learning better features**
>
> We thank Reviewer WB96 for this insightful suggestion regarding "general-purpose" versus "specialized" pretraining. Based on STAMP’s design principles, we firmly believe that incorporating non-cancer tissue is critical for learning robust and generalizable features:
>
> **1. Importance for Tokenization (Establishing a “Normal” Baseline):**
> Our abnormal gene ranking tokenization strategy relies on a global mean gene expression calculated across all 35 organs and multiple health states. The robustness of this mean depends on the normal baseline provided by non-cancerous and healthy tissues. If we were to train only on cancer data, this mean would be severely distorted, causing the word “abnormal” to lose its meaning. Non-cancer data is essential for the model to understand what is truly an anomaly (i.e., cancer).
>
> **2. Importance for Downstream Generalizability (Learning Common Structures):**
> Our model needs to perform well on both non-cancer tasks (like DLPFC) and cancer tasks (like HBC). Non-cancer data (e.g., healthy kidney, diseased liver, healthy brain) teaches the model to recognize common, fundamental biological tissue structures. These typical structures (such as stroma, blood vessels, and immune cells) are also abundantly present in the Tumor Microenvironment (TME) of cancer WSIs. If trained only on cancer data, the model would not learn to understand these critical TME components well, which would likely harm its performance on cancer tasks.
>
> Thus, non-cancer tissue is an indispensable part of our training data, contributing directly to STAMP's robustness as a general-purpose model. We have added this discussion to the Conclusion and Discussion section (Future Work) to clarify the theoretical importance of data diversity. We will also prioritize the empirical ablation in future work.

---

> ### Author Response · Authors · 2025-11-21
> **Response to Reviewer WB96 (5/5)**
>
> ### **8. Regarding the rationale for selecting DLPFC and the model's positioning (cancer-focused vs. general-purpose)**
>
> We thank Reviewer WB96 for this core question. We explicitly position STAMP as a **general-purpose model**, not "cancer-focused." This generalizability is intrinsic to our design:
>
> 1. **Methodological Generalizability (Tokenization):**
> Our “abnormal gene ranking” strategy relies on a global mean gene expression calculated from SpaVis-6M, which spans multiple tissues (35 organs) and multiple health states (including healthy, diseased, and cancerous). Therefore, an “abnormal” gene does not explicitly mean “cancer.” It can also refer to a brain-specific gene, as its expression is abnormal relative to the global average of all 35 organs. This makes our token sequence a powerful “identity ID” that can recognize both disease states (like cancer) and tissue types (like the brain).
>
> 2. **Training Data Generalizability:**
> Our core alignment dataset is a biologically balanced cohort in which non-cancerous tissue (56.9%/56.6%) forms the slight majority (including diseased, healthy, and treated tissues).
>
> 3. **Evaluation Benchmark Generalizability (Downstream tasks):**
> To validate this generalizability, our six downstream evaluation datasets are also balanced: three cancer-related datasets (HBC, HER2+, LUAD-mutation) and three non-cancer datasets (DLPFC (healthy), PSC (diseased), HHK (healthy)).
>
> **Rationale for Selecting DLPFC:**
>
> This is precisely why we chose DLPFC (a non-cancer dataset) to lead our evaluation. We selected DLPFC based on three key considerations:
>
> 1. **Necessity:** ST datasets with fine-grained spot-level annotations are incredibly scarce, and DLPFC is one of the few publicly available datasets that provides high-quality, manually annotated ground truth for cortical layers.
> 2. **Standardization:** It is one of the most commonly used and highly recognized benchmarks in the spatial transcriptomics field for evaluating clustering and alignment.
> 3. **Challenge:** It is the perfect platform for testing the effectiveness of our multimodal fusion. As discussed in Section 4.2 of the main text, DLPFC’s unique challenge is its subtle visual differences, which cause pure-vision models to perform poorly (see Table 1).
>
> Consequently, leading with DLPFC was a strategic choice: If we had only tested on HBC (a cancer dataset with clear visual features), we would not have been able to prove that our molecular supervision signal was truly playing a key role. STAMP’s SOTA performance on DLPFC (Table 1) conclusively demonstrates that our model (via its general-purpose tokenization) successfully identified the non-cancerous brain tissue types and fused the visual and molecular information.

---

> ### Comment · Area_Chair_Lj8F · 2025-11-27
> **Please engage in the discussion with authors**
>
> Please engage in the discussion with authors, thank you.
>
> AC

---

### Official Review · Reviewer_H1Wd · 2025-11-09

**Soundness:** 3
**Presentation:** 3
**Contribution:** 2
**Rating:** 6
**Confidence:** 5

**Summary:**

This work introduces STAMP, a framework for pathology-ST pretraining. STAMP works by aligning image patches and spatially resolved transcriptomic profiles via variety of reconstruction and contrastive learning objectives. This work also introduces SpaVis‑6M, a new dataset using 10X Visium with 1982 slices and 5.75m spots. STAMP was benchmarked on a variety of tasks including linear probe (DLPFC,  HBC), gene expression prediction (PSC, HHK, and HER2+) and LUAD mutation prediction (as well as other tasks) and compared against many pathology image encoders and ST prediction models.

**Strengths:**

- This is a good resource and benchmark paper for researchers working in pathology-genomics pretraining. SpaVis‑6M is a good contribution, as well as the pretrained weights and code used to pretrain on this dataset.
- Experimental design is overall strong, with diverse breadth of tasks and comparisons. Table 6 and Table 7 are important ablations, which respectively show (1) loss objectives used in STAMP improve multimodal pretraining performance, (2) performance gain of pretraining on SpaVis-6M versus HEST-1k.
- Figures are illustrative (Figure 5-9), with lots of good details included in the Appendix.

**Weaknesses:**

- What is the statistical significance of STAMP improvement? In Table 2, the standard deviation of performance is enormous for the PSC, HHK, HER2+ tasks. For the HER2+ task, most models have an average MSE of ~0.9 with a standard deviation of ~0.45.
- Is there a reason why STAMP was not evaluated on the HEST benchmark?
- There are many works looking at multimodal alignment of pathology and ST. While STAMP was compared against BLEEP and mclSTExp, it is missing many other comparisons such as HisToGene, Hist2ST, UMPIRE, and OmiCLIP, with OmiCLIP being the most relevant comparison that this work should compare against.
- It is not clear how all the different variations of STAMP are implemented (STAMP-V, STAMP-G, STAMP-F, STAMP-Reg, STAMP-Con). Are these all different models, or the same model but using the outputs of different heads and intermediate layers as the learned representation? Having a section on implementation details on how all the STAMP variants are set-up would be helpful.

This work makes a lot of good contributions to the biomedical community. However, the contributions are more empirical and less relevant for the fundamental ML/AI community. The most important contributions I see in this work is the introduction of the SpaVis-6M dataset and experimental design. Method-wise, the added loss components are mostly incremental but the authors still show they are helpful. While I think this paper would be a better fit for a more biomedical venue, I still lean towards acceptance.

**Questions:**

See above.

---

> ### Author Response · Authors · 2025-11-21
> **Respone to Reviewer H1Wd (1/3)**
>
> ### **Overall Reply to Reviewer H1Wd**
>
> We sincerely thank Reviewer H1Wd for their professional evaluation and insightful comments on our manuscript. We are very pleased that the reviewer recognized the contribution of the SpaVis-6M dataset, our strong experimental design, and the essential ablation studies.
>
> We have carefully considered all feedback and have addressed the five concerns in our detailed responses.
>
> ---
>
> ### **1. Regarding the high standard deviation in Table 2**
>
> Thank you for flagging this. The large SDs in the gene-expression prediction task reflect data heterogeneity, not instability of STAMP.
>
> 1. **Data heterogeneity.** Variance concentrates in HHK and HER2+: HHK spans six healthy donors, HER2+ spans seven patients., Patient-level CV amplifies inter-patient batch effects, yielding higher fold-to-fold variance.
> 2. **Task design.** We predict the top 5,000 HVGs—by definition, the most variable genes—so cross-patient/region prediction is inherently noisy.
>
> This pattern holds across all methods (STNet, EGN, His2ST, BLEEP, etc.), indicating a dataset/task property rather than a model-specific issue.
>
> Despite the variance, STAMP leads in mean performance across datasets and metrics. On HER2+, for example, STAMP$_{Con}$ attains PCC-V 0.279 and PCC-E 0.266, the best among the compared approaches.
>
> ---
>
> ### **2. Regarding the evaluation on the HEST-Benchmark**
>
> To clarify our evaluation protocol, we distinguish the HEST collection—which served as our pretraining data source—from the HEST-Benchmark. Crucially, we enforced a strict separation: any slide used in our downstream evaluation (e.g., DLPFC, HBC) was explicitly removed from the HEST collection before training. This guarantees that our downstream tasks are actual Out-of-Distribution (OOD) tests.
>
> We did not adopt the HEST-Benchmark for two reasons:
>
> 1. **Platform mismatch.** STAMP and SpaVis-6M are designed for 10x Visium (spot-level). The HEST-Benchmark is dominated by Xenium data (five of nine tasks), which has subcellular resolution and a different data structure and noise profile.
> 2. **Overlap with pretraining.** The benchmark’s four Visium tasks (55 slides) make up ≈16.2 % of our 339-slide alignment set. Removing them to avoid leakage would weaken pretraining; keeping them would invalidate evaluation.
>
> Instead, we use six public datasets that better match STAMP’s scope and allow a fuller assessment of generalization:
>
> 1. **Clustering / fine-grained classification:**
>   **DLPFC** is the standard ST clustering benchmark with subtle visual cues, ideal for testing how molecular supervision enhances discrimination.
>   **HBC** offers clear region boundaries, letting us verify that adding molecular input does not harm performance when visual features already suffice.
> 2. **Large-scale gene regression:**
>   **PSC** is BLEEP’s original benchmark, enabling direct comparison on the same task.
>   **HHK** contains six slices from six donors, capturing strong inter-patient variability and testing robustness to batch effects.
>   **HER2+** uses the Spatial Transcriptomics platform, providing a true cross-platform evaluation from Visium to another technology.
> 3. **WSI-level classification:**
>   **LUAD** evaluates whether features pretrained with spatial-gene guidance generalize to slide-level prediction.
>
> In summary, we omitted the HEST-Benchmark due to Visium–Xenium incompatibility and data overlap, and instead chose an OOD, Visium-aligned, and task-diverse suite that more rigorously tests STAMP’s generalization and cross-platform strength.

---

> ### Author Response · Authors · 2025-11-21
> **Respone to Reviewer H1Wd (2/3)**
>
> ### **3. Regarding the comparison to OmiCLIP**
>
> We thank Reviewer H1Wd for suggesting OmiCLIP. Note that His2ST is already listed in Table 2. We have now evaluated OmiCLIP on our benchmarks and incorporated the results into Tables 1 and 2 of the new PDF.
>
> 1. **New Experimental Results for OmiCLIP:**
> We have added OmiCLIP to Table 1 and Table 2 (see new PDF). These latest results clearly reveal the value of different supervisory signals:
>
> * **On Table 1 (Classification and Clustering):** OmiCLIP’s balanced acc was only 0.381, which is lower than PLIP (0.429), and even lower than CLIP (0.415), and shows a considerable gap with our STAMP$_V$ (0.624). We speculate this is due to OmiCLIP’s core design: it only uses the names of the Top 50 highly expressed genes (Top 50 HEG names) as the supervisory signal from the ST side. We believe this supervisory signal contains information that is too sparse and insufficient to provide adequate supervision for the vision encoder, and may even have harmed its performance on complex visual tasks.
> * **On Table 2 (Gene Expression Prediction):** OmiCLIP’s performance was more competitive, as its task is more consistent with its "gene-association" pretraining. However, it was still significantly worse than our STAMP$_{reg}$ on most metrics (e.g., on HER2+, MSE 1.110 vs 0.904).
>
> This comparison highlights the true advantages of STAMP again:
>
> * Our **supervisory signal** (the "Top 1500 gene ranks") is much richer, enabling STAMP$_V$ to achieve SOTA performance on Table 1.
> * Our **complete multimodal model (STAMP$_{con}$)**, empowered by spatially-aware designs like $\mathcal{L}_{CGR}$, remains the all-around SOTA on Table 2, outperforming OmiCLIP on all metrics.
>
> 2. **Discussion regarding UMPIRE / HisToGene:**
> Finally, regarding the UMPIRE and HisToGene models mentioned by the reviewer, we agree they are relevant works in this field. We have added a detailed technical discussion of them to the "Related Work" (Section 2) in the new PDF, clearly articulating the key advances STAMP makes upon these works.
>
> ---
>
> ### **4. Clarifying STAMP variants**
>
> We thank Reviewer H1Wd for flagging this. These variants are not separate models; they are different outputs/modes of a single trained STAMP used for downstream tasks. Our definitions were scattered in the original submission:
>
> * **STAMP$_V$ (Vision).** Features from the aligned vision encoder. Used in Table 1 for unimodal clustering/classification.
> * **STAMP$_G$ (Gene).** Features from the aligned gene encoder. Used in Table 1.
> * **STAMP$_F$ (Fused).** Concatenation of STAMP$_V$ and STAMP$_G$ features. Used in Table 1.
> * **STAMP$_{Reg}$ (Regression).** Frozen STAMP$_V$ plus a linear regression head for per-spot gene prediction; vision-only baseline in Table 2.
> * **STAMP$_{Con}$ (Contrastive).** Full model with vision + gene encoders, using the query–reference strategy for gene prediction; multimodal setting in Table 2.
>
> We agree that the separation was confusing. In the new PDF, we centralize definitions: add **STAMP$_V$ STAMP$_G$ STAMP$_F$** at the start of Section 4.2, and place **STAMP$_{Reg}$** and **STAMP$_{Con}$** at the beginning of Section 4.3.

---

> ### Author Response · Authors · 2025-11-21
> **Respone to Reviewer H1Wd (3/3)**
>
> ### **5. On the methodological contributions**
>
> We thank Reviewer H1Wd for recognizing our empirical results. While our application is biomedical, STAMP addresses fundamental challenges in learning from high-dimensional, sparse, and spatially correlated multimodal data. We believe our methodological contributions offer valuable insights to the broader ML/AI community beyond the specific domain of spatial transcriptomics:
>
> 1. **Robust Representation for High-Dimensional Sparse Data (Novel Tokenization):** Standard continuous regression or naive discretization fails on data with extreme sparsity and long-tail distributions (common in genomics, user-item interactions in RecSys, or text). Our Rank-based Tokenization introduces a novel inductive bias that prioritizes relative order over absolute values. This offers a generalizable strategy for denoising and robustly embedding high-dimensional sparse vectors, relevant to researchers working on robust representation learning.
> 2. **Topology-Aware Self-Supervision (Graph-Guided Masking):**
> Standard Masked Image Modeling (MIM) or MLM operates on regular grids or sequences. Our Contextual Gene Reconstruction ($\mathcal{L}_{CGR}$) extends this to irregular, non-grid graph structures. By enforcing consistency between a node and its topological neighborhood, we provide a generic framework for Graph-based Self-Supervised Learning. This is applicable to other spatially correlated non-grid data, such as geospatial sensor networks or 3D point clouds.
> 3. **Hierarchical Multimodal Alignment (Cross-Scale Mechanism):**
> Aligning modalities with inherent scale discrepancies (e.g., global image context vs. local molecular signals) is a fundamental challenge in Multimodal Learning. Our Cross-Scale Patch Positioning ($\mathcal{L}_{CSP}$) pretext task explicitly models the hierarchical relationship between local fine-grained features and global context. This "zooming" mechanism offers a blueprint for hierarchical alignment tasks in computer vision, such as aligning satellite imagery with ground-level attributes or matching video clips to long-form text summaries.
>
> In summary, STAMP is not just a solution for Visium data but a framework exploring how to fuse **pixels (dense, structured)** with **genes (sparse, high-dimensional, graph-structured)**. We believe these architectural choices provide a valuable reference for the ML community tackling complex, multi-structured multimodal fusion.

---

> ### Comment · Area_Chair_Lj8F · 2025-11-27
> **Please engage in the discussion with authors**
>
> Please engage in the discussion with authors, thank you.
>
> AC

---

### Author Response · Authors · 2025-11-21
**General Response: Sincere Thanks and Summary of Revisions**

We sincerely thank all the reviewers for their thoughtful and detailed feedback, as well as the time and effort devoted to reviewing our manuscript. We are encouraged by the positive recognition of our **SpaVis-6M dataset**, **methodological novelty**, and **strong empirical results**.

We have carefully addressed each reviewer's comments and suggestions. Their valuable insights have significantly improved the quality and clarity of our work. **We have uploaded the revised manuscript, and to make the revisions clear, we have highlighted the updated sections in red.**

---

### **Key Revisions and Highlights:**

**1. Rigorous Baseline Comparisons (Response to H1Wd & WB96):**
* We have supplemented our experiments with **OmiCLIP** results in Table 1 and Table 2 to provide a more comprehensive benchmark against state-of-the-art models.
* To address concerns about domain shifts in mutation prediction, we conducted additional experiments on **FFPE datasets**, where STAMP achieved SOTA performance on 6 out of 8 metrics (see Response to WB96).

**2. Enhanced Interpretability (Response to pd5B):**
* We added **attention heatmap visualizations** for WSI classification tasks (LUAD-mutation) in **Appendix A.5.3**.
* Comparative analysis reveals that STAMP generates **spatially coherent** attention maps that precisely localize viable tumor regions, effectively correcting the "attention drift" and noise observed in the uni-modal baseline (UNI).

**3. Methodological Clarifications (Response to ALtH & WB96):**
* We clarified our **Gene Tokenization** strategy (Section 3.2), explicitly stating how our rank-based approach handles data sparsity and avoids zero-imputation artifacts.
* We refined the rationale for selecting **UNI** as the vision backbone (Section 3.3 & Appendix A.3.4), explaining how this choice avoids the **semantic bias** of image-text models and the **global aggregation bias** of bulk-RNA models.

**4. Expanded Metrics and Analysis:**
* We supplemented **Accuracy (ACC)** metrics for WSI classification tasks (UBC-OCEAN, TCGA-NSCLC, PANDA) in Appendix A.5.3.
* We moved the **Limitations** discussion to the main text (Section 5) and expanded the **Future Work** section to emphasize the theoretical importance of **data diversity** (i.e., including non-cancerous tissues).

For other specific concerns, please refer to our detailed point-by-point responses to your individual comments.

---

### Author Response · Authors · 2025-12-01
**Summary of Rebuttal Status (Avg Score: 7.0, finalized pre-leak) & Guide to Revisions**

**To the newly assigned Area Chair:**

We understand the significant workload you are facing. To assist you, we provide a concise summary of the reviewer consensus reached before the system freeze, and a pointer to our detailed revision log.

**1. Current Status & Consensus (Timeline Verified):**
Following the rebuttal interactions, our submission achieved an **average score of 7.0 (Scores: 8, 8, 6, 6)**, with all reviewers leaning towards acceptance.

**Crucially, the score increase (from Reviewer ALtH) and the consensus were finalized on November 24, 2025.** This precedes the public disclosure of the OpenReview bug (Nov 27), ensuring these evaluations reflect genuine scientific consensus unaffected by recent irregularities.

* **Reviewer ALtH (Nov 24):** **Increased score from 6 to 8**. Explicitly stated: *"Most of my concerns seem cleared. I will increase the score!"*
* **Reviewer pd5B (Nov 24):** **Maintained score of 8**. Confirmed: *"The authors have addressed all my concerns."*
* **Reviewer H1Wd & WB96:** Rated 6 (Positive). We provided detailed responses covering all their questions.

**2. Summary of Revisions:**
To respect your time and avoid redundancy, we will not repeat the full details here. Please refer to our **"General Response: Sincere Thanks and Summary of Revisions"** (posted on **Nov 21, 2025**) for the comprehensive list of changes.

We hope this summary helps you navigate the review history efficiently.

Sincerely,
The Authors

---

### Meta-Review · Area_Chair_JDbK · 2025-12-24

**Summary:**

There is consensus that the paper makes an empirical contribution to computational pathology through both data and a model. First, the construction of  SpaVis-6M , the largest Visium-based spatial transcriptomics dataset to date is highlighted as a solid contribution, alongside the release of pretrained weights and code. Reviewers broadly agree that the experimental design is well executed with informative ablation studies. In adddition, the work introduces STAMP, a multimodal foundation-model pretraining framework that jointly learns representations for histopathology images and spatially resolved gene expression. It uses spatial transcriptomics (ST) as a grounded supervisory signal to guide visual representation learning. STAMP is claimed to be task-agnostic - i.e. the pretrained encoders can be reused for diverse downstream tasks such as gene expression prediction, spatial clustering, and whole-slide biomarker classification. It consists of two main encoders : (1) a vision encoder on multiscale image patches, and (2) a gene encoder that models gene expression with spatial awareness. The gene encoder is pretrained using masked gene modeling and spatial context objectives to capture both co-expression structure and neighborhood dependencies across tissue spots.

My overall reading of this work is one of introducing a new dataset as well as a simple, yet single model that achieves strong results on this data. Given the importance of spatial transcriptomics in the biomedical community, I think the data and this model as a baseline will be a good addition to the literature.

**Reviewer Concerns:**

The rebuttal reframes contributions as a general methodology with some useful tools but these are adaptations of existing paradigms that limits the novelty of the proposed model. However, given the dataset and the evaluations done on the model I think this concern is limited.

**Reviewer Scores:**

Atleast one reviewer states that they would increase the score.

---

### Decision · Program_Chairs · 2026-01-26

Accept (Poster)